# Characterization and clustering of kinase isoform expression in metastatic melanoma

**David O. Holland**[1], **Valer Gotea**[1], **Kevin Fedkenheuer**[1], **Sushil K. Jaiswal**[1], **Catherine Baugher**[1], **Hua Tan**[1], **Michael Fedkenheuer**[2], **Laura Elnitski**[1]*

**1** Translational and Functional Genomics Branch, National Human Genome Research Institute, National Institutes of Health, Bethesda, Maryland, United States of America, **2** Lymphocyte Nuclear Biology, National Institute of Arthritis and Musculoskeletal and Skin Diseases, National Institutes of Health, Bethesda, Maryland, United States of America

* elnitski@nih.gov

**Data Availability Statement:** All relevant data are within the manuscript and its Supporting Information files.

## Abstract

Mutations to the human kinome are known to play causal roles in cancer. The kinome regulates numerous cell processes including growth, proliferation, differentiation, and apoptosis. In addition to aberrant expression, aberrant alternative splicing of cancer-driver genes is receiving increased attention as it could lead to loss or gain of functional domains, altering a kinase's downstream impact. The present study quantifies changes in gene expression and isoform ratios in the kinome of metastatic melanoma cells relative to primary tumors. We contrast 538 total kinases and 3,040 known kinase isoforms between 103 primary tumor and 367 metastatic samples from The Cancer Genome Atlas (TCGA). We find strong evidence of differential expression (DE) at the gene level in 123 kinases (23%). Additionally, of the 468 kinases with alternative isoforms, 60 (13%) had significant difference in isoform ratios (DIR). Notably, DE and DIR have little correlation; for instance, although DE highlights enrichment in receptor tyrosine kinases (RTKs), DIR identifies altered splicing in non-receptor tyrosine kinases (nRTKs). Using exon junction mapping, we identify five examples of splicing events favored in metastatic samples. We demonstrate differential apoptosis and protein localization between *SLK* isoforms in metastatic melanoma. We cluster isoform expression data and identify subgroups that correlate with genomic subtypes and anatomic tumor locations. Notably, distinct DE and DIR patterns separate samples with *BRAF* hotspot mutations and *(N/K/H)RAS* hotspot mutations, the latter of which lacks effective kinase inhibitor treatments. DE in *RAS* mutants concentrates in CMGC kinases (a group including cell cycle and splicing regulators) rather than RTKs as in *BRAF* mutants. Furthermore, isoforms in the *RAS* kinase subgroup show enrichment for cancer-related processes such as angiogenesis and cell migration. Our results reveal a new approach to therapeutic target identification and demonstrate how different mutational subtypes may respond differently to treatments highlighting possible new driver events in cancer.

**Funding:** Funding provided to LE by the Intramural program of the National Human Genome Research Institute, National Institutes of Health. grant number 1ZIAHG200323. https://www.genome.gov The sponsors or funders did not play any role in the study design, data collection and analysis, decision to publish, or preparation of the manuscript.

**Competing interests:** The authors have declared that no competing interests exist.

## Author summary

The incidence of melanoma continues to rise worldwide, especially in young adult populations. Metastasized melanoma is difficult to treat with current drugs and may kill patients in a matter of months. Kinase inhibitor (KI) drugs have shown success in treating melanoma with $BRAF^{V600E}$ mutations, and a better understanding of how melanoma alters the human kinome may reveal new drug targets. We use two approaches: finding kinases with altered gene expression and finding kinases with aberrant alternative splicing, which is less studied. Alternative splicing is a mechanism through which a gene may produce different gene products (isoforms), even if overall gene expression does not change. We find multiple examples of aberrant splicing and discuss their possible role in driving cancer. Because melanoma cells lacking a *BRAF* mutation do not respond well to current KIs, we also contrast results between genomic subtypes of melanomas. In particular, samples with a primary *NRAS* mutation have distinct expression patterns. Our results buoy future research interests of oncologists; and will also be of interest to researchers studying aberrant splicing in other diseases. Additionally, we provide novel algorithms for statistical testing of altered isoform makeup of transcripts generated from the same gene locus.

## Introduction

Melanoma is the deadliest form of skin cancer, with about 232,100 new cases and 55,500 deaths worldwide each year [1]. Although incidence is less than 5% of new cancer cases in the U.S., incidence and deaths worldwide continue to rise, especially in the young adult populations [2]. Stage 1 or 2 disease is easily treated by surgery, where 5-year survival rates are > 90% [1], but if not caught early tumors may metastasize to the nearby lymph nodes and then throughout the body. Once the disease reaches the brain, median survival time decreases to 5 months [3]. Thus, novel systemic treatments for metastatic melanoma are needed.

Kinases have become compelling cancer targets because they contain mutations that produce constitutive kinase activation and dysregulate signaling pathways in cancer. Among the 538 known kinase genes in humans, there are numerous relevant targets. Specifically, mutations have been observed in kinases serving as growth factor receptors [4], cell cycle regulators [5,6], nuclear signaling [7], and apoptosis regulators [8]. In melanomas, *BRAF* is most commonly mutated, along with other kinases including *NRAS* and *NF1*. Fleuren et al. identified 23 additional kinases harboring driver mutations for melanoma, including the receptor *FGFR3* and cell cycle regulator *CDK4* [9]. Additional targets may remain undiscovered as atypical kinases, which can phosphorylate proteins but lack a typical kinase domain.

Along with chemotherapy and immunotherapy, treatments for advanced melanomas also incorporate small molecule kinase inhibitors (KI). There are currently 37 FDA approved KIs on the market for cancer treatment, with ~150 in ongoing clinical trials [10]. Targets of these small molecule KIs include *BRAF*, which occurs in about 50% of melanoma patients [1,11], and *MEK*, a downstream signaling target of *BRAF* in the *MAPK* pathway. Despite initial successes for these drugs, limitations remain. For example, half of all *BRAF*-mutant tumors treated with *BRAF* inhibitors advance within 6–8 months post-treatment [12] whereas other hotspot mutations, such as in *NRAS*, lack effective KI treatments altogether [13]. Complementary targeted approaches in the form of immune-checkpoint blockers ipilimumab, pembrolizumab, and nivolumab, have recently been shown to significantly improve survival in some patients, even in those with wildtype *BRAF* [14–16]. Although these treatments do not work in

the majority of patients [17], combining them with KIs may improve survival prospects. Thus while existing drugs show promise for a subset of patients, new targets and combination therapies are in dire need to address treatment-resistant tumors, and especially those tumors with wildtype *BRAF*.

There are multiple forms of kinase dysregulation: activating mutations, overexpression, underexpression, copy number alterations, repression, and chimeric translocations; but there has been much less research into gene isoform distributions, in part due to the difficulty of estimating isoform composition from short read RNA sequences [18,19]. For these data, computational approaches are required to estimate isoform counts prompting development of transcript alignment algorithms such as *RSEM* [20], and faster pseudo-alignment algorithms such as *kallisto* [21]. The gold-standard of isoform analysis might eventually be achieved through "3$^{rd}$ generation" long read sequencing technologies such as PacBio [22] and Oxford Nanopore [23], providing more accurate, contiguous isoform sequences, although these currently have a high error rate and are costly compared to 2$^{nd}$ gen. sequencing [24]. Regardless, long and short read sequencing technologies both discern differential isoform composition to address the question of how alterations in sequential exon continuity can change functional outcomes.

Although isoform distributions are not widely reported in the literature, there is reason to suspect they are altered in cancer tissues. First, alternative splicing is highly abundant under normal conditions where up to 94% of human genes undergo alternative splicing [25], and the dominant isoform depends on cell type [26]. Second, in various cancers, trans-acting splicing factors can be mutated or mis-regulated [27–31], potentially skewing isoform distributions. Third, somatic DNA mutations–abundant in cancer–may occur on splice sites, favoring or suppressing splicing events. Kinases are known to undergo alternative splicing events in cancer [18] and these are implicated in tumor progression. Examples include *MKNK2* in glioblastoma [32]; *CD44* in breast cancer [33]; and *KLF6* in prostate, lung, and ovarian cancers [34]. Splicing induces losses or gains of functional or regulatory domains, documented in cancers, altering the functions of affected proteins in the cell. Despite these observations, differential isoform usage is an extra level of detail not normally analyzed in cancer studies.

Here we propose to detect and demonstrate the biological relevance of isoform alterations in metastatic melanoma. Notably, a recent study of the human kinome in prostate cancer found that there was little overlap between genes with differential expression and genes with differential splicing [35], suggesting a study of the latter will yield additional therapeutic targets. Despite our emphasis on differential isoform expression, we include differential expression of genes (i.e., representing a gene locus with a single expression value), to show distinct and relevant findings learned from each type of assessment.

In this study, we analyze RNA-seq data from The Cancer Genome Atlas (TCGA) skin cutaneous melanoma project (SKCM) to study changes to the kinome of metastatic vs. primary tumor melanomas. Important findings include isoforms downregulated in metastatic samples that correspond with known and novel suppressors of metastasis and additional subgroupings of metastatic samples with narrowly focused therapeutic potential. Our results identify characteristics of wildtype *BRAF* tumors, as well as new subdivisions among *BRAF* mutant tumors.

## Methods

### Human kinome

We obtained Gene IDs for 538 human kinases from the Human Kinome database [36] at http://kinase.com/web/current/kinbase/. Kinases are classified into 10 phylogenetic groups: tyrosine kinases (TKs); "Sterile" serine/threonine kinases (STEs); calmodulin-dependent

kinases (CAMKs); Cdk, MAPK, GSK, and Cdk-like related kinases (CMGCs); protein kinase A, protein kinase G, and protein kinase C related kinases (AGCs); tyrosine kinase-like (TKLs); casein kinase 1 (CK1); receptor guanylate cyclases (RGCs); "other" typical kinases; and atypical kinases (aPKs) i.e. kinases that phosphorylate without a conventional kinase domain; For our analysis, we further subdivided TKs into receptor tyrosine kinases (RTKs) and non-receptor tyrosine kinases (nRTKs) due to their distinct functional roles.

## TCGA data

We obtained RNA-seq data and kinase gene counts–estimated using *HTSeq* [37]–from the National Cancer Institute (NCI)'s Genomic Data Commons (GDC) portal for TCGA's skin cutaneous melanoma (SKCM) project. This included data from 472 samples gathered from 468 patients: 367 samples for metastatic tumors, 103 for primary tumors, 1 for an additional metastatic tumor from the same patient, and 1 for solid normal tissue. The latter two samples were not used in our analysis.

The data was processed in 14 batches, with the largest batch (labeled "A18") having 218 of the samples in three plates. The remaining batches had 10–48 samples in a single plate each.

## Isoform quantification

For the purpose of quantifying the abundance of isoforms in the human kinome, we used the *kallisto* (v0.45.0) package [21] in conjunction with the transcript sequences of protein coding genes in the Gencode (release 29) annotation of the human genome. We first constructed the *kallisto* index file using the 98,913 FASTA sequences of transcript isoforms of human protein coding genes included in the Gencode annotation (ftp://ftp.ebi.ac.uk/pub/databases/gencode/Gencode_human/release_29/gencode.v29.pc_transcripts.fa.gz; accessed March 15, 2019). FASTQ-formatted RNA-Seq reads (48-bp, paired-end) for each TCGA SKCM sample were produced from the bam files obtained from the Genomics Data Commons Data Portal. In order to avoid biases in *kallisto* estimates of fragment lengths, for each sample we produced FASTQ files in which the order of the reads was randomized. We then used these randomized reads to perform the *kallisto* "quant" analysis, from which we obtained the transcripts per million (tpm) estimates of each isoform abundance.

## Sample quality control

3' bias for each sample was estimated using the QoRTs package [38]. For sample purity, we used the consensus purity estimate from Aran et al. [39]. Samples with purity $< 70\%$ were removed to create our "high purity" sample set. Samples with a QoRTs 3' bias score $> 0.55$ (see ref [38] for Methods) were also removed in our "quality controlled" set. After clustering kinase isoform expression in metastatic samples, we also classified 83 metastatic samples as having amounts of immune infiltrate using k-means clustering with 2 centers (see Clustering of Metastatic Samples below).

## Differential expression (DE)

We tested differential expression of all genes between primary tumor and metastatic samples using the *DESeq2* toolbox for R [40] with two models: "sample type" and "sample type + batch" to account for batch effects.

Genomic subtypes for 56 primary tumor and 260 metastatic samples were obtained from Akbani et al. [11]. They included *BRAF* hotspot mutants (47%), *RAS [N/H/K]* hotspot mutants

(29%), *NF1* mutants (9%), and triple wildtype (WT) (15%). The remaining 156 samples were added after the study and had no genomic subtype information.

## Reverse phase protein array (RPPA) data

The Reverse Phase Protein Array (RPPA) level 3 normalized data were downloaded from the GDAC data portal (http://gdac.broadinstitute.org/). The original data contains 355 SKCM samples consisting of 92 primary tumor and 263 metastatic samples. Since the RPPA data used antibodies in rabbit and mice, we manually mapped the protein names into human gene names, with the aid of GeneCards (https://www.genecards.org/). We found 165 unique genes corresponding to the 208 RPPA protein probes. This included 33 kinase genes with 56 (26.9%) corresponding probes. We focused on the 224 samples with purity $\geq$70%, same as in our differential gene expression analysis. We tested differential protein expression between primary tumor (n = 78) and metastatic samples (n = 146) using a two-sided Wilcoxon's rank sum test. Benjamini-Hochberg adjusted p-value < 0.05 was deemed significant.

## Calculations for differential isoform ratios (DIR)

Transcript isoform counts for the TCGA samples were estimated from RNA-seq data with *kallisto* [21], using isoform information for protein coding loci provided by Gencode v.29 transcriptome annotation. In total, there were 3,040 protein coding isoforms for the human kinome. 69 genes with only one coding isoform and one pseudogene in the kinome list (*PRKY*) were not tested, leaving 2,971 isoforms. For each gene, isoform counts (in transcripts-per-million or TPM) were grouped as a vector (e.g. a five-element vector for a gene with five coding isoforms), and the vector was normalized to sum to 1. One vector per sample was made, ignoring samples with zero counts for all isoforms.

We used two models to test for differential isoform ratios. The first was a permutation method utilizing linear discrimination analysis (LDA). LDA was performed to reduce the space of isoform vectors to the 1D line which best separates sample types, and the LDA statistic

$$\frac{\left(\mu_{PT} - \mu_{Met}\right)^2}{\sigma_{PT}^2 + \sigma_{Met}^2}$$

was calculated. The sample labels were then randomized $n_{iter}$ times and the statistic recalculated to create a null distribution, from which the p-value was found. This method had the benefit of producing a single p-value without assumptions, but could only find p-values as low as $1/n_{iter}$.

In the second model, principal component analysis (PCA) was performed on the space of normalized isoform vectors, providing us with "n-1" components for "n" isoforms. PCs with zero variance were removed. We tested the difference in isoform coordinates between sample types along each PC using one of three different statistical tests (see below) and combined the p-values using Fisher's method. For both models, p-values were adjusted using Benjamini-Hochberg FDR adjustment.

## Comparison of statistical tests

Given that the permutation test becomes computationally prohibitive for large datasets and high precision, we attempted to find a statistical test that could reproduced the results obtained through permutations. We used three different tests along the principal components of the space of isoform vectors: the Wilcoxon rank sum test, Welch's t-test, and the general

independence test from R's conditional inference (coin) package [41]. We combined the p-values from each principal component with both Fisher's method (FM) and the asymptotically exact harmonic mean (HMP) from DJ Wilson [42]. This resulted in six sets of p-values which we compared to the permutation test results.

We found that the t-test combined with Fisher's method gave the best correlation between p-values (r = 0.92) and ranks (ρ = 0.92), while the coin test combined with HMP gave the best correlation between the logarithm of p-values (r = 0.89). However, total correlation may be of less interest than the sensitivity and specificity of the tests. We calculated Youden's J statistic (sensitivity + specificity– 1) at three significance levels: p = 0.05, 0.01, and 0.001. The t-test combined with Fisher's method performed best at all three levels, with J = 0.79, 0.80, and 0.80 respectively, followed by the coin test with Fisher's method. The geometric mean of these two tests $p_{new} = \sqrt{p_{t-test}p_{coin}}$ performed better, with J = 0.80, 0.84, and 0.86 respectively, and also increased all three correlations. We thus adopted this test for scaling up the number of genes. The Wilcoxon test performed poorly due to difficulties handling ties in the data.

## Clustering of metastatic samples

A quasi-Poisson generalized linear model (GLM) was used to test each individual metastatic sample vs. all primary tumor samples for each protein-coding isoform–using TPM counts from *kallisto*–resulting into a 3,040 x 367 matrix of p-values. Before clustering the data was thresholded into three bins, setting all p < 0.05 to +1 for isoforms with increased expression, all p < 0.20 to -1 for isoforms with decreased expression, and all other entries to 0. The reason we used such a liberal p-value for negative change is because most count data follow a Poisson-like distribution with a low median, which makes decreased expression for individual samples unlikely to test as significant. For example, isoform SLK-202 tests as highly significant for decreased expression (p = 3.4e-9) for all metastatic vs. primary tumor samples but only tests as significant (p = 0.0014 and 0.038) for two individual samples.

After digitizing, we applied k-means clustering to the data matrix, using the elbow method to find an appropriate number of clusters. Enrichment for tumor region, mutation subtype [11], batch ID, and kinase phylogenetic group in each cluster were tested using Fisher's exact test.

## Gene biological process and kinase group enrichment

1,572 biological process (BP) annotations were downloaded from the PANTHER database at geneontology.org. Genes were ranked by p-values (for DE or DIR) and significant genes tested for enrichment using the one-sided Fisher's exact test (i.e. hypergeometric test), using the remaining kinase genes as the background. We found that enrichments could differ drastically depending on the p-value threshold chosen for significance, so we searched for BP enrichment at multiple thresholds. Additionally, testing for DE or DIR with small sample sizes produced less extreme p-values than testing with large sample sizes, resulting in comparing >300 significant genes from one set of results (more than half the kinome) to <10 genes in another set of results. So we tested four percentile-based thresholds–the top 5%, 10%, 20% and 40% of all genes with a p-value–to obtain a comparable set of enrichments between sample sets. Results described are for the top 5% of genes unless noted otherwise.

We did not adjust p-values for the biological processes for several reasons. Having discovered a set of significant genes, we wanted to investigate the functional role served by these genes. Some annotations, such as "protein kinase", will never test as significant because all the genes in our background and foreground are kinases, making the expected false discovery rate lower than assumed in Benjamini-Hochberg (BH) correction. Furthermore, GO terms are

highly dependent, making common adjustment methods such as BH inappropriate. Finally, GO terms do not account for individual isoform activities, thus do not address our underlying question.

We did calculate an empirical false discovery rate (see S1 Results) merely to compare our enrichment results to those of a randomly selected set of "significant" genes.

Kinase phylogenetic group enrichment (see "Human kinome" above) was calculated in the same manner using percentile thresholds, with p-values unadjusted.

## Split-read alignment mapping

To evaluate changes in the relative abundance of isoform using an alternative method, we quantified the relative abundance of split reads specifically associated with the isoform of interest. For this purpose, we aligned the RNA-Seq reads using STAR against the hg19 version of the human genome assembly. We used the QoRTs package [38] to quantify split read support for splice junctions. For cases of alternative promoters, we compared the relative abundance of split reads supporting a common exon junction with alternative upstream exons. In cases of isoforms differentiated by a skipped exon, we considered the reads supporting the junction skipping the exon, and the average number of reads supporting the two junctions of the alternatively spliced exon. The relative abundance was expressed as a fraction of reads specifically supporting one isoform out of the total number of reads supporting both isoforms. The difference in the relative abundance was compared between primary tumor and metastatic samples using a one-sided Wilcoxon rank sum test, guided by the expectation set by the output from the *kallisto* tool.

In addition to this method, we also performed local analysis of exon usage using the package *DEXSeq* [43] on the quality-controlled sample set (see "Sample quality control" above). All kinase genes, including those with only one coding isoform, were tested.

## Survival analysis

We obtained patient survival data, i.e. days until death, from TCGA. To determine differences in survival across sample clusters (see "Clustering of metastatic samples" above), survival events and their respective times up to 4000 days were compiled for samples in each cluster based on vital status. We then used this data to generate a Kaplan-Meier estimator to plot the survival curves of each cluster. Log-rank tests were used to evaluate significance.

We assessed the correlation between kinase gene expression and patient survival using overall survival calculated for 205 high purity metastatic samples with survival data. For each kinase gene (n = 538), *HTSeq* gene counts (normalized by size factor by *DESeq2*) were correlated against overall survival using the Spearman correlation test. P-values were adjusted using Benjamini-Hochberg method.

## Overexpression of SLK isoforms in Metastatic Melanoma

We transiently overexpressed short-length *SLK* (*SLK-201*), and full-length *SLK* (*SLK-202*) in the metastatic melanoma cell line A375 (ATCC, Manassas, VA). SLK-201 and SLK-202 were cloned into the GFP fusion expression vector, p-RECEIVER-M98 (Genecopoeia, Rockville, MD). A375 cells were grown in Dulbecco's Modified Eagle Medium (DMEM) with 10% FBS. A375 cells were transfected with lipofectamine 2000 (Invitrogen, Carlsbad, CA). Deletions $\Delta_{1-373}SLK$-201 and $\Delta_{1-373}SLK$-202 were made from the full-length constructs by GenScript (Piscataway, NJ).

To determine early apoptosis, the cells were stained with annexin V (BD Biosciences, San Jose, CA), and analyzed by FACS at 24h, 48h, and 72h time points. The positive threshold for

annexin V detection was determined by comparing a negative control (cells treated with the same volume lipofectamine used in transfection) and a positive control (cells treated with 1 μM of Adriamycin, a DNA damaging drug which induces apoptosis) at each time point for each replicate. Similarly, the positive threshold for GFP expression was determined by comparing a negative control (cells treated with a volume lipofectamine used in transfection, but no vector) and a positive control (cells transfected with eGFP-only vector) at each time point for each replicate. We analyzed percent annexin V in GFP positive cells over time for the negative control (i.e., Lipofectamine, no vector), eGFP-only, *SLK*-201-eGFP, *SLK*-202-eGFP. Cells expressing GFP were binned into five groups of increasing GFP fluorescence intensity for further analysis as follows: B1 ($10^4$–$10^{4.5}$), B2 ($10^{4.5}$–$10^5$), B3 ($10^5$–$10^{5.5}$), B4 ($10^{5.5}$–$10^6$), B5 ($> 10^6$).

To examine differences in the actin cytoskeleton, the cells were stained with Phalloidin-iFluor 594 (Abcam, Waltham, MA) and DAPI (Thermo Scientific, Waltham, MA). They were visualized with a Zeiss LSM 880 NLO Laser Scanning Microscope at 24h, 48h, and 72h time points.

## Results

We analyzed the 538 kinase genes comprising the human kinome for changes in total mRNA expression and 3,040 isoforms for altered isoform expression, between metastatic and primary tumors. Using computational tools *HTSeq* [37] and *kallisto* [21] with short read sequences, we implemented the data analysis workflow depicted in Fig 1. Along with differential expression defined at the gene level and differential isoform ratios calculated within each locus, we performed a clustering analysis to identify pathway, mutational and functional characteristics that define each subgroup.

In this paper, we will first cover the DE results for varying sample sets (all samples, high purity samples only, and samples separated by genomic subtype), covering significant genes and their biological process enrichments. We will then do the same for the differential isoform ratio results before comparing the two groups.

### Sample demographics

Primary (n = 103) and metastatic (n = 367) tumors were obtained from the TCGA skin cutaneous melanoma project (SKCM) (Table 1). Primary tumors originated in a number of locations

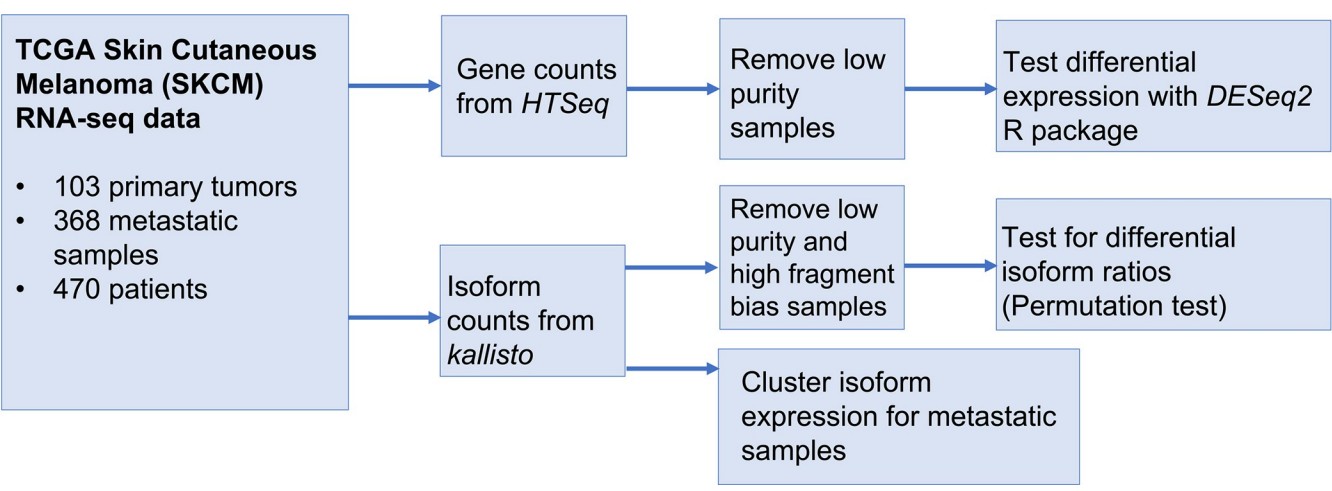

**Fig 1. Data analysis workflow.**

**Table 1. Sample demographics.**

| Sample details | Variables | Sample numbers | | |
|---|---|---|---|---|
| type | - | Normal tissue | Primary | Metastatic |
| count | - | 1 | 103 | 367 |
| tumor origin | arms or legs | - | 41 | 153 |
| | trunk | 1 | 48 | 125 |
| | head or neck | - | 8 | 30 |
| | other (armpit, genitalia, etc.) | - | 4 | 9 |
| | unknown | - | 2 | 50 |
| metastatic location | regional cutaneous or subcutaneous tissue | - | - | 74 |
| | regional lymph nodes | - | - | 221 |
| | distant metastases | - | - | 68 |
| | unclassified metastases | - | - | 4 |
| genomic subtype | *BRAF* hotspot mutation | - | 32 | 118 |
| | *RAS* hotspot mutation | - | 11 | 81 |
| | *NF1* any mutation | - | 5 | 23 |
| | triple WT | - | 8 | 38 |
| | not available | - | 47 | 107 |
| sex | male | 1 | 61 | 230 |
| | female | - | 42 | 137 |
| age (median) | - | 51 | 65 | 56 |
| race | white | 1 | 94 | 353 |
| | Asian | - | 7 | 5 |
| | non-white Hispanic | - | 1 | 2 |
| | black | - | 0 | 1 |
| | unknown | - | 1 | 6 |
| # of batches | | 1 | 14 | 14 |

including arms or legs, trunk, head or neck, or other areas, such as armpit, genitalia, etc. Metastatic locations included regional cutaneous or subcutaneous tissue, regional lymph nodes, distant metastases, and unclassified metastases. Samples were skewed towards males, and mostly derived from white individuals. Patient age at time of diagnosis ranged from 15 to 90, with a median of 58-years-old.

## Differential expression (DE) dominated by receptor tyrosine kinases

We first tested differential expression at the gene level. Out of 538 kinase genes, 281 (52%) had significant DE ($p_{adj} < 0.05$) between all primary tumor and all metastatic samples (S1 Table). The top groups, ranked by p-value, included both non-receptor (nRTKs) and receptor tyrosine kinases (RTKs) (Fig 2A). We looked for biological process enrichment in the top 5% and 10% of genes, and found strong enrichment for immune cell activation (both innate and adaptive). Clustering analysis (see Methods) revealed these genes have strongly correlated expression, suggesting their high expression results from immune infiltrate in the metastatic samples, i.e. immune cells attacking tumor cells. Using this approach, we identified 83 metastatic samples with high amounts of putative immune infiltrate (see Methods), which we removed before rerunning the *DESeq2* analysis. This action removed the enrichment for nRTKs, whereas RTK enrichment remained (Fig 2B). We next addressed the impact of impure tumor samples, as measured by the consensus purity estimate developed in Aran et al. [39]. When 168 samples with < 70% estimated purity were removed from the original set, which included 80 of the 83

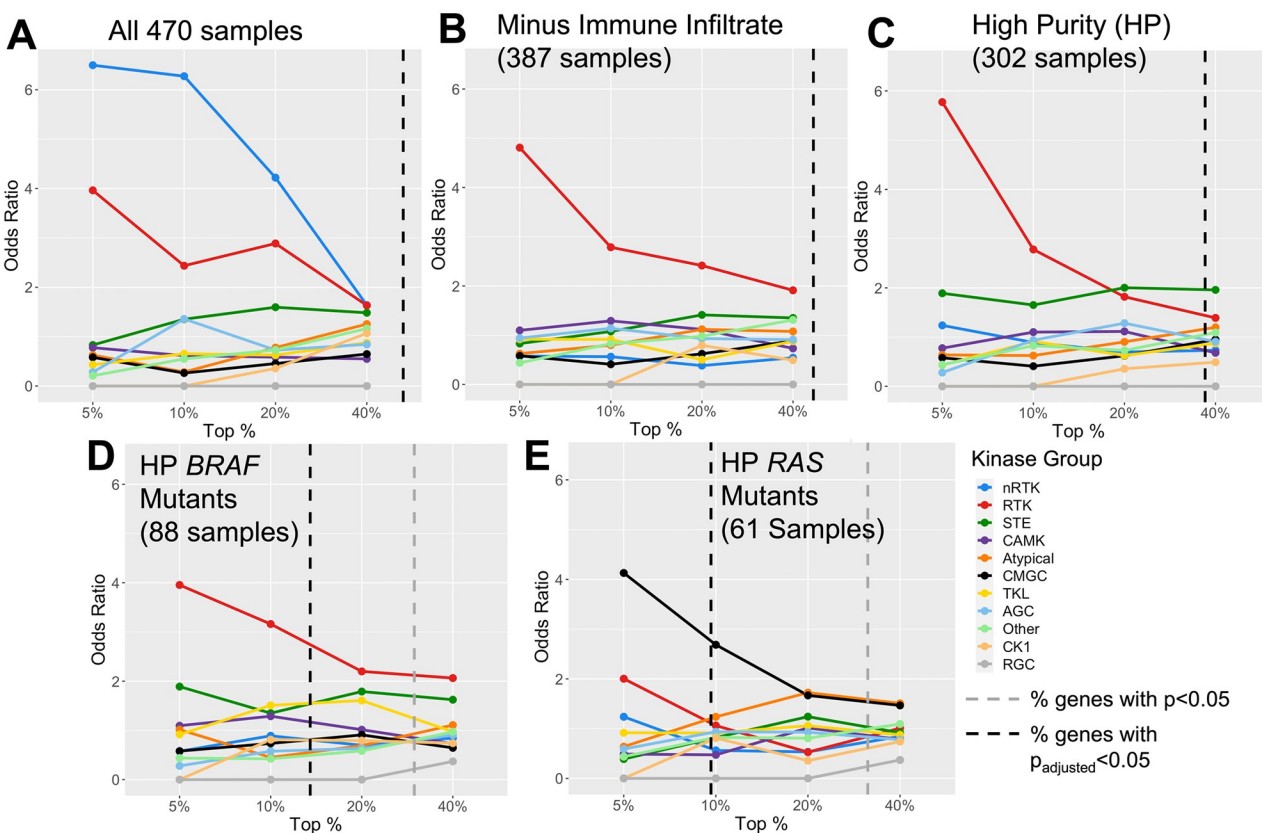

**Fig 2. Kinase group enrichment for differential expression of primary and metastatic tumors differs by sample set.** Depicted are the odds ratios for each kinase group in the top 5%, 10%, 20%, and 40% of kinase genes, ranked by p-value. This indicates that the strongest DE enrichment is concentrated in nRTKs for all 470 samples, RTKs for high purity and BRAF mutant samples, and CMGC kinases for RAS mutant samples. Enrichment data collected at the four percentile points are independent of p-value; the percent of genes that have significant DE (p<0.05) before and after p-value adjustment are shown by the gray and black dotted lines respectively. Sample type (primary tumor or metastatic) was the only model variable for the DESeq2 results.

immune infiltrate samples, again we saw enrichment for nRTKs disappear whereas RTKs remained significant (Fig 2C). We named this filtered group the "high purity" (HP) group. In both assessments, enrichment for RTKs remained significant when assessed as subsets of the top 5% to ~ 20% of genes. There was also a lesser enrichment for the STE kinase group (p = 0.031 at 20% threshold; Fig 2C), which contains kinases upstream of MAPK signaling cascades. When only the high purity (HP) samples were compared we found 197 significant genes including 26 of the 57 RTKs (S2 Table). Of these 26 kinases, 11 are known to be internalized from the cell surface into the nucleus [44]: *FGFR1/3, FLT1, ERBB4, INSR, TIE1, CSF1R, EGFR, IGF1R, MET*, and *KDR*. Internalized receptors have been linked to cancer progression and resistance to therapy by, for example, activating DNA damage response pathways [45–47]. Absolute fold-changes for significant genes ranged from 0.503 (*KSR*) to 11.7 (*NRK*).

Next, we examined differential gene expression when the HP primary tumor and metastatic samples were subdivided into their particular genomic subtypes (*BRAF, RAS*, triple WT and *NF1*) (S3 Table). Although this approach reduced sample size for each test, a similar enrichment pattern emerged. Specifically, the DE genes for the *BRAF* hotspot mutants, *NF1* mutants, and triple WT samples were all enriched for RTKs (odds ratios = 4.0, 6.9, and 3.2 respectively at 5% threshold: Fig 2D). The deviant result was the case of the *RAS* hotspot mutants, where DE was dominated not by RTKs but by CMGC kinases (odds = 4.1, Fig 2E). This group

**Table 2. Differential expression from primary tumor to metastatic samples in receptor tyrosine kinases (RTKs).**

| Gene Name | Base Mean | Fold Change | P-value | P-adj | Nuclear Trafficked [44] |
|---|---|---|---|---|---|
| EPHA1 | 149.44 | -6.13 | 8.15e-31 | 1.82e-28 | |
| FGFR3 | 496.72 | -4.58 | 4.29e-14 | 3.72e-12 | Yes |
| LMTK3 | 84.69 | -2.99 | 4.87e-10 | 2.97e-08 | |
| EPHA3 | 671.47 | 3.20 | 7.27e-09 | 3.77e-07 | Yes |
| EPHB6 | 392.00 | -2.79 | 1.31e-07 | 5.62e-06 | |
| FLT1 | 1539.25 | 1.57 | 4.10e-06 | 1.24e-04 | |
| MERTK | 993.10 | 1.86 | 2.16e-05 | 5.06e-04 | |
| ROR1 | 409.93 | 1.75 | 4.58e-05 | 9.44e-04 | |
| FGFR2 | 232.15 | -2.54 | 6.50e-05 | 1.25e-03 | Yes |
| EGFR | 731.80 | -1.95 | 3.58e-04 | 4.76e-03 | Yes |
| EPHA2 | 3375.91 | -1.56 | 1.74e-03 | 0.0156 | |
| TIE1 | 895.76 | 1.42 | 3.63e-03 | 0.0267 | Yes |
| EPHA6 | 16.09 | 2.24 | 5.71e-03 | 0.0371 | |
| DDR2 | 6622.61 | 1.36 | 6.66e-03 | 0.0416 | |
| RYK | 2858.81 | 1.17 | 8.31e-03 | 0.0484 | Yes |
| INSR | 2475.26 | 1.25 | 8.60e-03 | 0.0495 | Yes |

Results are between high purity (>70%) primary and metastatic tumors using sample type and batch ID as model variables. Only RTKs with $p_{adj} < 0.05$ are shown. Negative fold change indicates decreased expression in metastatic samples.

contains both cyclin-dependent kinases–which regulate the cell cycle–and downstream MAP-kinases–which regulate gene expression–as well as kinases directly involved in splicing regulation (i.e., serine arginine protein kinases). Although RTKs (particularly Ephrin receptors, i.e., *EPHA*) remain significantly altered in the RAS mutants, this result suggests a distinct set of alterations is associated with metastases in RAS mutants. In metastatic *BRAF* mutants, mutated *BRAF* itself had non-significant increased expression.

## Influence of sample batches on differential gene expression

Because not all the samples in the TCGA data set came from the same batch, we also ran *DESeq2* using both sample type and batch ID as model variables. This approach increased p-values, decreasing the number of significant genes. However, 123 kinase genes remained significant at the $p_{adj} < 0.05$ level, including 16 RTKs (Table 2), compared to 197 total genes when only the sample type was the variable. Gene ranking was not substantially altered (Spearman correlation, $\rho = 0.81$) and enrichment trends were similar to our prior results for all genomic subtypes (*BRAF*, *RAS*, triple WT), with the exception of the *NF1* mutant samples. These could not be assessed due to the small sample size (2 primary and 11 metastatic tumors), where the primary tumors and metastatic samples were not from the same batch. Excepting this subtype, for the remaining analyses we included batch ID as a model variable.

## Biological process (BP) enrichment differs by genomic subtype

In addition to the kinase group enrichment, for each set of results (i.e., all samples, high purity, *BRAF* mutants, etc.) we looked for BP enrichment among significant genes, as a hypothesis-free approach to further characterize the metastatic tumors (S4 Table). Top genes were highly enriched for immune-related annotations when all 470 samples were used, the highest being "adaptive immune system" (p = 5.1e-9) (Table 3). These enrichments nearly disappeared when samples with < 70% purity were removed. Surprisingly, when only analyzing HP samples,

**Table 3. Summary of kinase differential expression results.**

| Sample Set | Sample Size | P<0.05[1] | P$_{adj}$<0.05 | Kinase Group Enrichment | Selected BP Enrichments[2] | P-value[2] |
|---|---|---|---|---|---|---|
| All Samples | PT: 103<br>Met: 367 | 262 genes | 202 genes | nRTK<br>RTK | Adaptive immune response<br>Hemopoiesis<br>Innate immune response | 5.1e-9<br>1.4e-7<br>6.1e-4 |
| All Samples (Purity>0.7) | PT: 88<br>Met: 214 | 203 genes | 123 genes | RTK<br>STE | Ephrin receptor signaling pathway | 0.033 |
| *BRAF* Hotspot Mutation (Purity>0.7) | PT: 27<br>Met: 61 | 99 genes | 27 genes | RTK | Cell differentiation<br>Positive reg. of lipase activity<br>Biomineralization<br>Positive reg. of neurogenesis<br>Positive reg. of cell projection organization<br>Ephrin receptor signaling pathway | 1.3e-4<br>5.2e-4<br>5.9e-4<br>0.0016<br>0.0058<br>0.0077 |
| *RAS* Hotspot Mutation (Purity>0.7) | PT: 9<br>Met: 52 | 133 genes | 36 genes | CMGC | Eye morphogenesis<br>Positive reg. of defense response<br>Reg. of angiogenesis<br>Ephrin receptor signaling pathway | 0.0034<br>0.017<br>0.050<br>0.035 |
| *NF1* Any Mutation (Purity>0.7) | PT: 2<br>Met: 11 | 62 genes | 12 genes | RTK | Reg. of MAPK cascade<br>Eye morphogenesis<br>Chemotaxis<br>Neuron projection guidance | 0.0054<br>0.0057<br>0.013<br>0.014 |
| Triple WT (Purity>0.7) | PT: 8<br>Met: 20 | 41 genes | 9 genes | RTK<br>CAMK | Calcium-mediated signaling<br>Cellular response to cytokine stimulus<br>Inflammatory Response<br>Defense Response | 0.0021<br>0.0031<br>0.0053<br>0.013 |

[1] Both sample type and batch ID were used as model variables for *DESeq2*, except for the *NF1* subtype where only sample type was used

[2] Enrichments are for the top 27 (5%) kinase genes ranked by p-value

significant BP annotations were depleted, with only four annotations receiving a p-value below 0.05. "Ephrin receptor signaling pathway" (p = 0.033); was the only non-immune-related enrichment. Significant ephrin pathway genes included 7 ephrin receptors and downstream non-RTKs such as ROCK1/2 (regulators of actin cytoskeleton, downstream of RHOA and EPHA4 [48]), PAK3 (downstream of RAC1 and EPHBs and important for cytoskeletal reorganization in dendritic spines [49]) and YES1 (oncogene downstream of EPHA2 which induces cell proliferation and migration [50]). Ephrin receptors are prototypical RTKs that impact cell shape, adhesion, and movement through activation or repression of the Rho GTPase family [51], suggesting an important role in metastatic processes.

The lack of BP enrichments suggests either that DE is widely distributed among a number of cell processes, or that enrichment patterns differ by genomic subtype and disappear when lumped together. To address this question, we separated the high purity samples into genomic subtypes and found support for the latter hypothesis, where division into individual subtypes revealed enrichment in distinct processes (Table 3). We observed strong BP enrichment among DE genes for samples with *BRAF* mutations, with the most significant annotation being "cell differentiation" (p = 1.3e-4). Neurogenesis and cell projection-related enrichments were also discovered. The DE genes for *RAS* mutants had weaker enrichments, although select examples such as "positive regulation of defense response" and "regulation of angiogenesis" are relevant for cancer. The ephrin receptor signaling pathway was enriched in both the *BRAF* (p = 0.008) and *RAS* (p = 0.035) mutants.

The *NF1* mutant and triple WT sets had smaller sample sizes (13 and 28 samples respectively). The *NF1* mutants were enriched for "regulation of MAPK cascade" (p = 0.0054), "chemotaxis", and "neuron projection guidance" among others. The triple WT samples–unlike the

other genomic subtypes–were enriched for responses to cytokine stimulation, especially interleukin-1 (p = 0.0053), as well as the inflammatory response and defense response.

## Reverse Protein Phase Array (RPPA) Data

We compared our results with an orthogonal dataset containing reverse protein phase array (RPPA) data. Although isoform information was not available, 33 kinase genes had available RPPA data, wherein 14 genes (42%) had significant ($p_{adj}<0.05$) differential expression (S1 Fig), compared to 60 of 175 non-kinase genes (34%) (S5 Table). The 14 genes include two RTKs, *ERBB3* and *KIT*. While the number of kinase genes covered by the RPPA data is too small for a signaling pathway enrichment analysis, a gene ontology analysis revealed that the 14 genes participate in vital biological processes related to cell growth and proliferation. In particular, the cell cycle regulatory genes including *EEF2K*, *PRKCD*, *PRS6KB1*, *CHEK2*, *MTOR*, and *BRAF* were all upregulated in the metastatic group. These results corroborate that some of the kinase genes are also dysregulated at the protein level as tumors progress from primary to metastatic state.

## Kinase genes exhibit differential isoform usage between primary and metastatic tumors

To complement the usual procedure of DE analysis, we next tested whether multi-isoform kinase genes exhibit differential isoform ratios (DIR) between primary and metastatic tumors. Per the Gencode v.29 annotation, we tested 468 such genes with 2,971 total coding isoforms. We measured significance in 317 (68%) via a permutation test ($p_{adj} < 0.05$) when all 470 tumor samples were used, more genes than had tested significant for DE. Our complementary PCA test (see Methods) found p-values as low as 5.3e-28 for *LIMK1*.

This high level of observed DIR could be an artefact of sample impurity–since different cell types might express isoforms in different ratios–or experimental artefacts such as fragment sequence bias [52]. Fragment bias results from degraded RNA reads. Because these reads are sequenced from the 3' end following poly(A) enrichment protocols, high levels of degradation results in overestimation of 3' fragment isoforms and underestimation of 5' fragment isoforms (Fig 3A and 3B), although the total gene count estimate is unaffected.

We inspected the isoform counts and found that genes with the strongest DIR had 3' fragment isoforms, suggesting samples with high 3' fragment bias could be driving significance. This bias was concentrated in the primary tumor samples (two-sided Wilcoxon, p = 1.3e-8) (Fig 3C). We also found sample impurity was concentrated in metastatic samples (p = 1.6e-4). Thus, both could contribute to the observed levels of significance.

Using the histograms as a guide, we removed samples with less than 70% purity or a QoRTs estimate of 3' bias > 0.55 from further analysis (Fig 3C). This reduced the number of samples to 50 primary tumor and 178 metastatic (S3 Table), which we deemed the "quality-controlled" (QC) sample set. In this stringent QC set, only 60 genes had DIR with $p_{adj} < 0.05$ (Permutation test), and the most significant kinase was *SLK* at p = 7e-6 (Table 4, full list in S6 Table). As a *post-hoc* analysis we tested the effects on individual genes by removing samples one-by-one to assess the influence of fragment bias or sample impurity. (See S1 Results, S2 and S3 Figs)

## Differential gene expression does not predict differential isoform ratios

Having controlled for fragment bias and impurity, we asked whether genes with differential expression between primary and metastatic tumors were also likely to exhibit DIR. We compared the p-values for DIR from the QC set to the p-values for DE from the HP set. Genes with $p_{adj} < 0.05$ had non-significant overlap (Fisher's exact test, p = 0.310), with only 15 genes

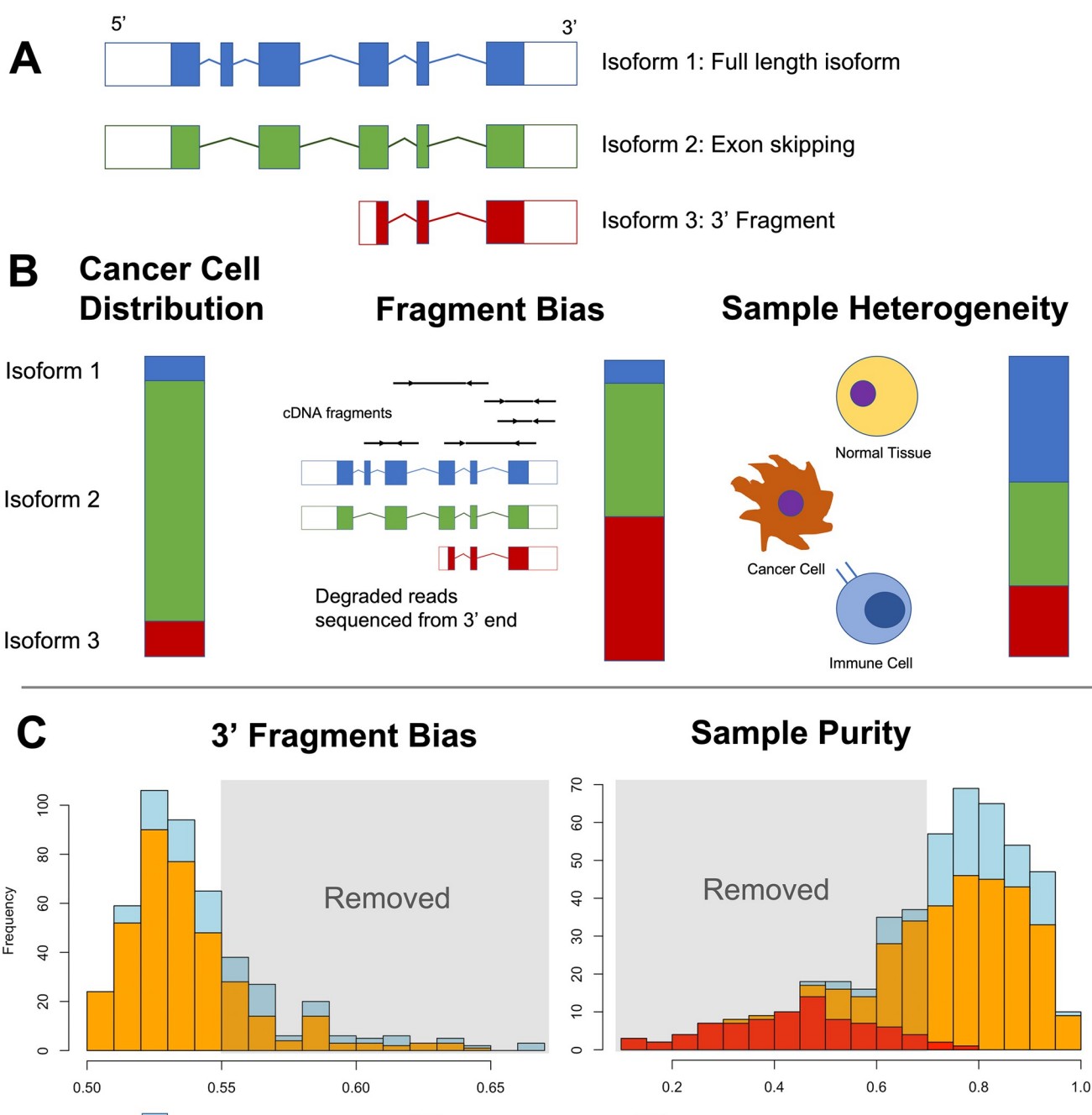

**Fig 3. Examining data bias in isoform count estimation.** (A) Isoform examples. (B) Cancer cell isoform distributions can be altered by effects of fragment bias and cell type heterogeneity (i.e. sample impurity). The bars represent the apparent relative expression level of each isoform, with the left bar indicating the true distribution in a cancer cell and the right two bars indicating how data bias can skew the results. (C) Primary and metastatic tumor samples assessed for 3' fragment bias using QoRTs, where samples scoring above 0.55 were removed, and sample impurity, where samples with < 70% purity were removed. Red bars in the sample purity assessment indicate metastatic samples with observed high immune infiltrate.

overlapping (Fig 4). Gene rank (by p-value) had no correlation (Spearman, ρ = 0.038). The gene *MAP3K3*, for example, had the third highest level of DIR (p = 5.6e-5) but no observed change in expression (p = 0.77, rank 487). Interestingly, genes with significant DIR were

**Table 4. Example genes with significant changes in isoform ratios after 3' bias and sample impurity filtering.**

| Gene | DIR, p-value | DIR, p$_{adj}$ | Coding Isoforms | Splicing/isoform changes in metastatic samples | Significant DE? |
|---|---|---|---|---|---|
| *SLK* | 7.00E-06 | 0.00328 | 2 | Skipping of 13th exon, part of coiled-coil region | **Yes ↑** |
| *TGFBR1[1]* | 3.40E-05 | 0.00590 | 9 | Selection for 3$^{rd}$ exon, encodes transmembrane domain | |
| *MAP3K3* | 5.60E-05 | 0.00590 | 5 | Skipping of 3$^{rd}$ exon, precedes PB1 domain | |
| *COQ8B[1]* | 6.80E-05 | 0.00590 | 11 | Selection for 6$^{th}$ exon, effects on function unknown | **Yes ↓** |
| *ABL1* | 8.00E-05 | 0.00590 | 3 | Increase of full-length isoforms, decrease of 5' fragment | |
| *LIMK1* | 0.000128 | 0.00750 | 3 | Alternate promoter site. Shortens 1$^{st}$ zinc- binding domain. | |
| *PAN3[1]* | 0.00027 | 0.0108 | 2 | Selection for 4$^{th}$ exon | |
| *FGFR3* | 0.00028 | 0.0108 | 7 | Switch to mutually exclusive version of 8$^{th}$ exon, affecting 3$^{rd}$ Ig-like domain | **Yes ↓** |
| *FES* | 0.00037 | 0.0108 | 9 | Skipping of 11$^{th}$ exon, encodes SH2 domain | |
| *UHMK1[1]* | 0.00067 | 0.0154 | 3 | Alternate promoter site favoring longer isoform with ATP-binding region | |
| *PAK6* | 0.00081 | 0.0173 | 14 | Decrease of all isoforms except a middle fragment | **Yes ↓** |
| *MAST4* | 0.00175 | 0.0256 | 16 | Potential alternate promoter (decreased use of first three exons of isoform -202) | **Yes[2] ↓** |
| *BLK* | 0.00215 | 0.0273 | 3 | Unequal increase of two major isoforms | **Yes ↑** |
| *MKNK2* | 0.0069 | 0.0531 | 8 | Alternate splicing at terminal exon, increase of isoform without MAPK binding site | **Yes ↓** |

[1]Results supported by *kallisto* counts but not supported by exon junction read mapping

[2]Significant DE only when using sample type + batch ID in model

enriched for nRTKs (Fig 5A) but not RTKs, the opposite of what we observed for DE genes. Thus DE and DIR affect different genes.

We separated the QC samples into genomic subtypes, as we did for the DE analysis, and calculated DIR for each subset. Due to the small sample sizes, few genes tested as significant with our permutation test. For example, the *BRAF* mutants revealed only four genes with p$_{adj}$ < 0.05 (*SLK*, *MOK*, *ABL2*, and *SYK*) while the other 3 subtypes revealed no significant genes after p-value adjustment (summarized in Table 5). As seen for the full sample set, no ranked gene list for any DIR sample group correlated with its DE counterpart.

## P < 0.05, Unadjusted

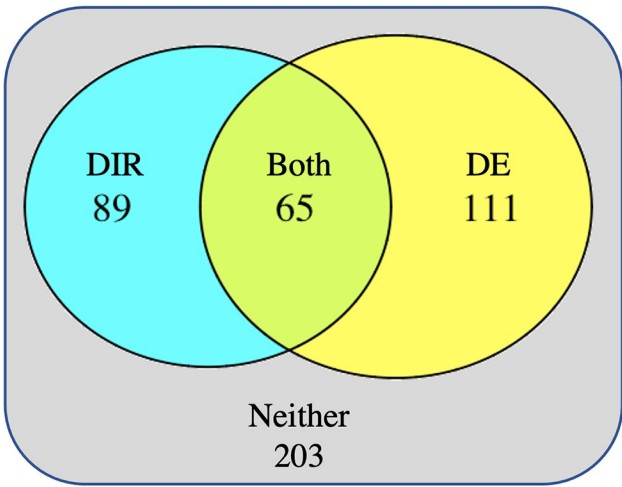

## P < 0.05, Adjusted

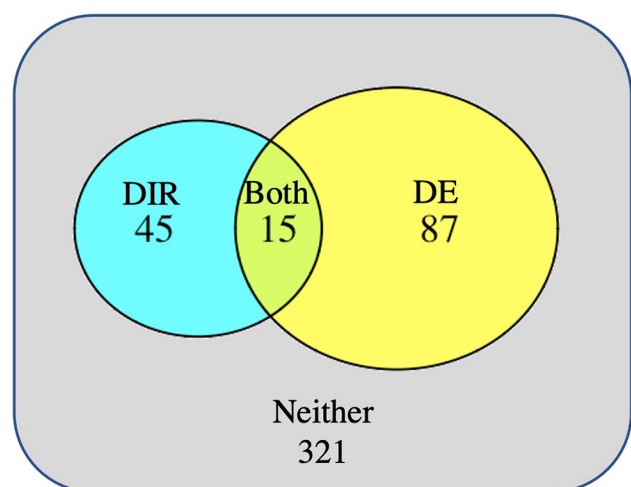

**Fig 4. DE does not overlap with DIR.** Significance in one does not predict significance in the other (One-sided Fisher's exact test, p = 0.091 (left) and 0.310 (right)). Compared are results using high purity samples for DE (model ~ sample type + batch ID) and quality-controlled samples for DIR. Compared are the 468 kinase genes with >1 coding isoform.

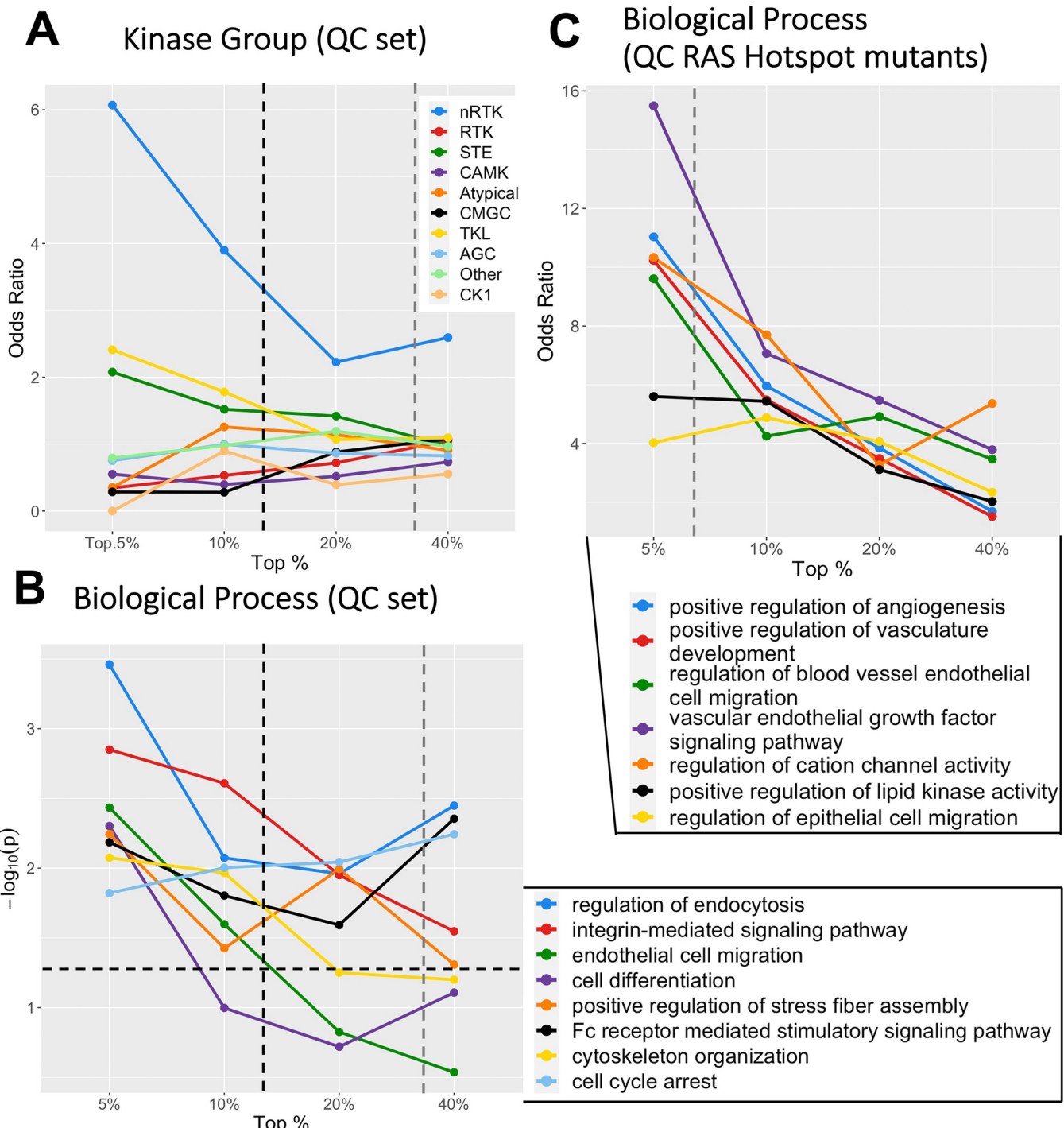

**Fig 5. Genes with significant DIR display unique BP enrichment patterns.** (A) DIR genes are enriched for non-receptor tyrosine kinases in the QC set, whereas there was no kinase group enrichment in the full sample set. The black dotted line indicates the percent of genes with $p_{adj} < 0.05$, and the gray dotted line the percent of genes with unadjusted $p < 0.05$ (B) Select biological process enrichments for the QC set. Note that significance is plotted ($-\log_{10}p$), not odds ratios, and the horizontal dotted line indicates $p = 0.05$. (C) When using the QC RAS hotspot mutant samples only, DIR genes were highly enriched for angiogenesis and related annotations.

**Table 5. Summary of kinase differential isoform ratio results.**

| Sample Set[1] | Sample Size | P<0.05 | P$_{adj}$<0.05 | Kinase Group Enrichment | Selected BP Enrichments[2] | P-value |
|---|---|---|---|---|---|---|
| All Samples | PT: 103 Met: 367 | 330 genes | 317 genes | None | Response to amino acid starvation Cytoskeletal organization Positive reg. of lipid kinase activity Response to fibroblast growth factor Blood vessel development[3] | 3.3e-4 4.5e-4 0.0021 0.0057 8.7e-4 |
| Quality Controlled | PT: 50 Met: 178 | 154 genes | 60 genes | nRTK | Reg. of endocytosis Endothelial cell migration Cell differentiation Positive reg. of stress fiber assembly Cytoskeletal organization | 3.4e-4 0.0037 0.0050 0.0057 0.0084 |
| *BRAF* Hotspot Mutation (QC) | PT: 17 Met: 52 | 57 genes | 4 genes | nRTK | Regulation of protein acetylation B cell receptor signaling pathway Reg. of cell motility | 0.0084 0.012 0.031 |
| *RAS* Hotspot Mutation (QC) | PT: 6 Met: 50 | 29 genes | 0 genes | None | Positive reg. of angiogenesis Cellular response to VEGF Chemotaxis | 1.6e-4 0.0020 0.0075 |
| Triple WT (QC) | PT: 6 Met: 16 | 40 genes | 0 genes | None | Reg. of gene expression Reg. of RNA splicing Chromatin organization | 0.0065 0.015 0.023 |

[1] Subgroups are comparable to Table 3, except the *NF1* mutant set, which had too few samples (1 primary tumor, 11 metastatic) for reasonable analysis.

[2] Enrichments are for the top 24 (5%) kinase genes with >1 coding isoform ranked by p-value.

[3] Blood vessel development is concentrated in the top 47 (10%) genes.

## DIR affects different biological processes than seen for DE

Because many unadjusted p-values were significant for DIR we elected to search for gene ontology (GO) enrichments. For each sample set, we searched for biological process (BP) enrichment in the top genes (ranked by p-value) using percentile thresholds from 5% - 40% (see Methods). Enrichments are described for the top 5% (24) genes unless noted otherwise.

For comparative purposes, we examined the full sample set first without filtering, which contained low purity and high fragment bias samples, we revealed 221 BP terms with p < 0.05 and 10 additional terms with p<0.001. The most significant terms included "positive regulation of translation" (p = 1.5e-4), "cytoskeletal organization", "response to amino acid starvation", and "blood vessel development" (Table 5). Immune-related enrichments were strongest at the 40% threshold, indicating putative immune infiltrate may affect DIR results, but the most significant genes were not immune-related.

The QC set had fewer BP enrichments than the full sample set (Table 5). These enrichments included "regulation of endocytosis" (p = 3.5e-4) "cytoskeletal organization", "endothelial cell migration", "cell differentiation", and "cell cycle arrest" (Fig 5B), all of which have a putative relevance to cancer.

The genomic subtype sets revealed distinct BP enrichments–as they did when testing DE genes. In contrast to the DE genes, the DIR genes between *BRAF* mutant primary and metastatic tumors did not show strong BP enrichments, while the DIR genes between *RAS* mutant samples showed enrichment for 94 BPs. The strongest of these was "positive regulation of angiogenesis" (p = 1.6e-4) and related enrichments such as "vasculature development". Other enrichments included "cell-cell communication", "protein transport", and "membrane organization" (Fig 5C). Such enrichment patterns would not be discovered if DE alone was studied. In contrast, significant genes from the *BRAF* mutants had 27 processes enriched below p = 0.05 –these included 6 cell locomotion-related enrichments–and none below p = 0.008.

### Resolving alternative splicing events in kinase genes

Focusing on DIR with discrete splicing changes, we identified skipped exons, alternative promoters, and alternative terminal exons (Table 4). For example, *ABL1* has two long isoforms (*ABL1-201* and *-202*), which differ only in their promoter site, that have increased expression in metastatic samples. An additional isoform *(ABL1-*203) encodes a shorter 5' fragment, and decreases in expression. However, *ABL1* does not test as significant in DE between primary and metastatic samples, indicating that the DIR analysis can reveal aberrations that differential gene expression does not capture.

To test the *kallisto* DIR data for evidence of splicing differences, we quantified RNA-seq reads mapped directly to the nucleotide sequences of exon junctions in several genes from Table 4. This provides a resolved view of exon splicing patterns in the samples which did not rely on *kallisto* (see Methods). Within the melanoma sequence data, we confirmed exon skipping in three genes–*MAP3K3* (exon 3), *FES* (exon 11) (Fig 6) and *SLK* (exon 13) (Fig 7). We also confirmed switching to mutually exclusive exons in two genes–exon 8 of *FGFR3* and the terminal exon of *MKNK2*–and increased use of an alternative promoter in *LIMK1* (Fig 6). We illustrate the fraction of split reads, out of all reads, supporting these events.

In *SLK*, the most significant gene on our list, expression of the long isoform *SLK-202* is decreased, whereas the short isoform *SLK-201* increases (Fig 7). The short isoform skips exon 13 predicting a putative role for loss of this exon in cancer. We compared expression of this alternative exon in normal melanocytes using RNA-seq data from Zhang et al. [53]. Exon 13 was absent in the normal cells, and largely specific to primary tumor samples.

Some genes have DIR which coincides with significant DE. For example, 6 of the 7 coding isoforms of *FGFR3* are suppressed in metastatic samples (S4 Fig), while the remaining isoform -205 has mildly increased expression. *PAK6*, with 14 isoforms, undergoes a similar alteration. In *BLK,* DIR of 3 isoforms is driven by an unequal increase of 2 major isoforms, rather than all 3.

To address the functional consequences and biological implications of isoform switching, we matched the alternatively spliced regions in these five genes to domain annotations obtained from UniProt (Table 4). The skipped exon in *SLK* encodes a section of a coiled-coil region in the C-terminal domain. *SLK* uses this domain to dimerize at high concentrations, and these dimers activate apoptosis [54]. *MKNK2* switches to a shortened terminal exon which lacks the MAPK binding site, interfering with downstream signaling. The 11[th] exon of *FES* encodes the SH2 domain, which is necessary to activate the kinase domain [55]. The 3[rd] exon of *MAP3K3* is not mapped to any domain, but it precedes the PB1 protein-interaction domain. These data indicate that the isoform changes modulate the usage of important domains in the kinases, which can ultimately affect their function and participation in signaling networks. Finally, the alternative promoter of *LIMK1* shortens the first zinc-binding domain, a domain that inhibit the protein's kinase activity [56].

### Comparison to *DEXSeq* results

In a parallel approach, we analyzed the primary and metastatic sample data using DEXSeq, a method commonly used to measure differential exon usage. *DEXSeq* found only 11 exonic bins in 5 genes to have significant differential usage ($p_{adj} < 0.05$), compared to 60 genes with our method (S7 Table). Three of these genes were also highly significant with our method: *MAST4*, *FGFR3*, and *SLK* (S5A–S5C Fig). The remaining two, *PDGFRA* and *LMTK3*, are likely false positives due to the low number of counts for their significant exons (median ~1) (S5E Fig). Before multiple test correction, the alternate promoter of *LIMK1* was significant (p = 0.0014); but not the SH2 domain of *FES*, MAPK-binding region of *MKNK2*, nor the 3[rd]

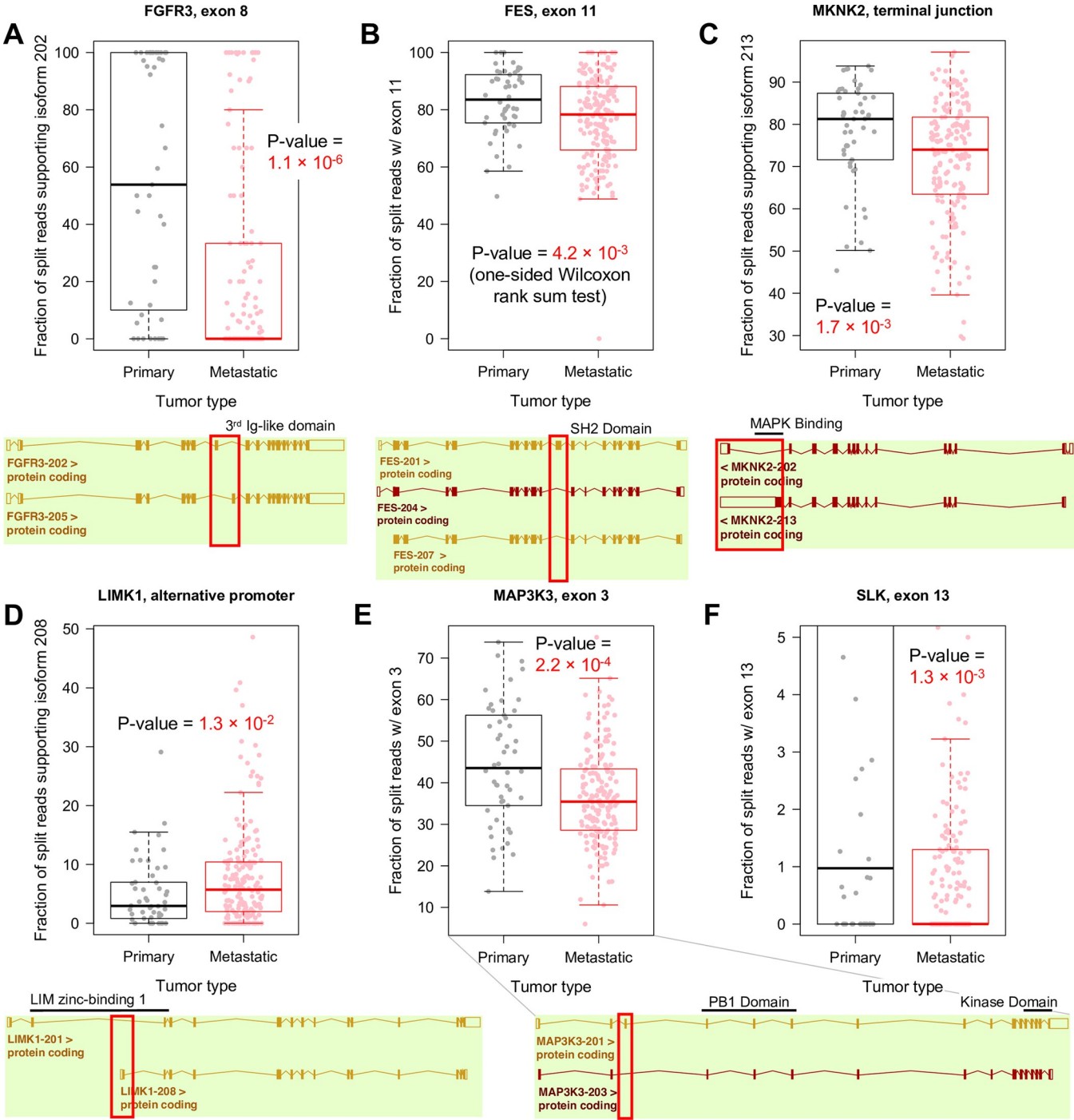

**Fig 6. Alternative splicing between primary and metastatic samples.** (A-E) Shown for each gene are box plots for fraction of split reads aligned to exon junctions in primary tumor and metastatic samples. Below, maps of each isoform identify the differential exon. (F) Zoomed-in plot for SLK. Half (89 / 178) of metastatic samples have no reads supporting exon 13. See Fig 7 for more detail.

exon of *MAP3K3*; despite our confirmation with direct junction sequence alignment. We found multiple reasons for the low sensitivity of *DEXSeq* (see S1 Results), which led to us to elect to measure DIR using isoform counts.

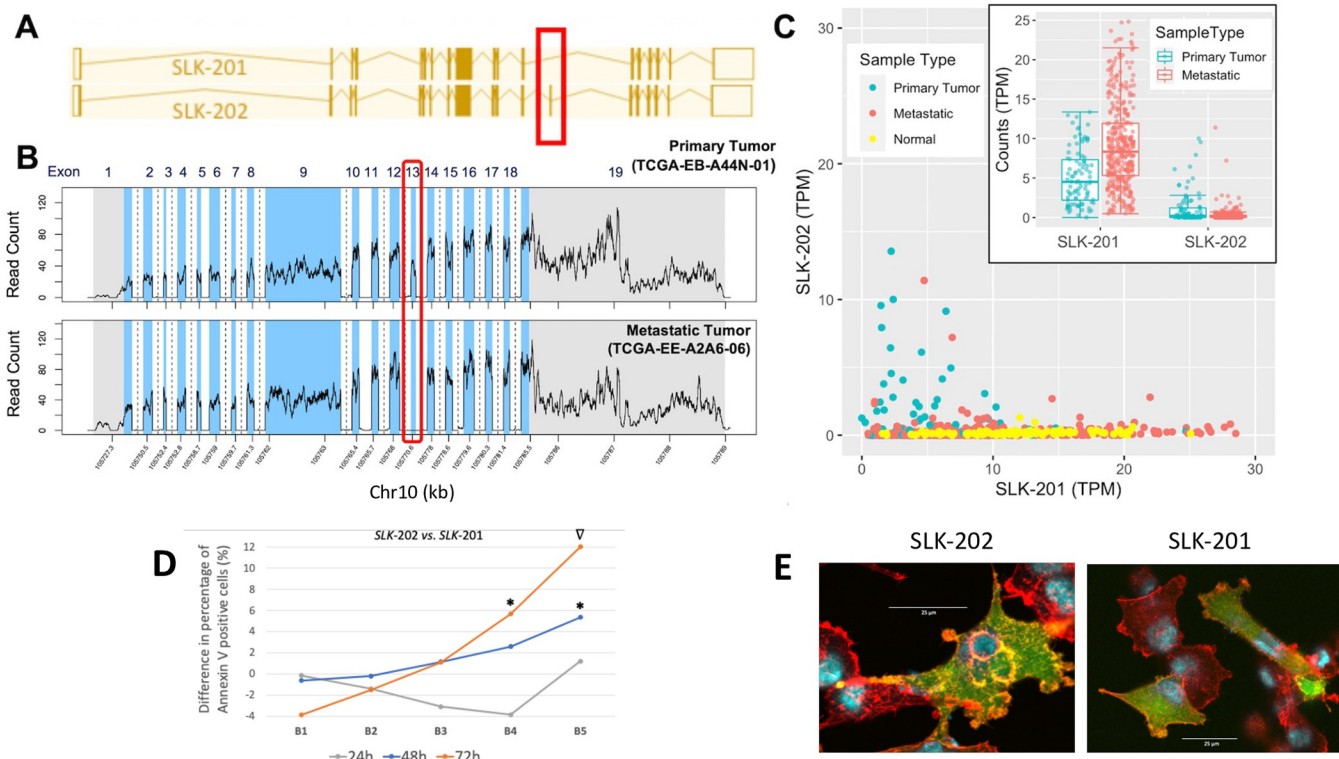

**Fig 7. Alternative splicing in *SLK*.** (A) The two coding isoforms of *SLK*, which only differ in the presence or absence of exon 13. (B) Mapped read counts from one primary tumor and one metastatic sample. In the metastatic sample, no reads are mapped to exon 13, indicating that isoform *SLK*-202 is not present; exons (blue), introns (white-dashed bar indicates introns longer than 300 bp, only partially shown), UTRs (gray). (C) Box and scatter plots for the two isoforms of *SLK*. Also shown in the scatter plot are the normal melanocyte samples from Zhang et al. [53]. (D) Difference in incidence of annexin V positive cells between *SLK*-202 (long) and *SLK*-201 (short) isoforms for bins, B1-B5, of increasing GFP fluorescence (see also S7A Fig) and different time points. Each value represents the average of two replicates (actual values and associated variance are provided in S7B–S7F Fig); *(p < 0.05), ∇(p = 0.055). (E) Merged channel images of A375 cells transfected with eGFP-only, *SLK*-202-eGFP, and *SLK*-201-eGFP constructs at the 24-hour timepoint. Cells were stained with DAPI (blue) and phalloidin (red), along with GFP expression (green). Scale bars = 25 μm.

## Overexpression of SLK isoforms in Metastatic Melanoma

*SLK* is involved in apoptosis and in the disassembly of actin [57]. We wished to see if overexpression of the two *SLK* isoforms could produce cell death in metastatic melanoma. We hypothesized that expression of the full-length isoform (*SLK-202*) would produce more cell death compared to the short-length isoform (*SLK-201*) due to the lack of one dimerization domain (coiled-coil region) in the shorter isoform. We also hypothesized that there would be differences in actin disassembly between SLK isoforms. In these experiments, *SLK*-201 and *SLK*-202 were cloned into p-RECEIVER-M98, an eGFP-fusion expression vector. We transiently transfected A375 metastatic melanoma cells with the negative control (i.e., Lipofectamine, no vector), eGFP-only, *SLK*-201-eGFP, and *SLK*-202-eGFP. Cells were collected at 24h, 48h, and 72h post transfection. We found no endogenous *SLK*-202 in A375 using RNA-seq data from the Sequence Read Archive (SRR961660; S6A Fig).

We observed a significant increase in the percentage of annexin V positive cells at the 48h and 72h time points in both *SLK* constructs compared to the eGFP-only control, supporting the hypothesis of *SLK*-induced apoptosis (S6B Fig). We observed no significant difference between the effect of the two *SLK* constructs when comparing all GFP positive, annexin V positive cells (S6B Fig). However, when eGFP expressing cells were divided into bins, B1–B5, corresponding to increasing expression levels of eGFP-fusion constructs (S7A Fig), we observed

an increase in the difference of percent annexin V positive cells between *SLK*-202 and *SLK*-201 at the 48h and 72h time points (Figs 7D and S7A). These data indicate that the long *SLK* isoform (*SLK*-202) induces apoptosis at a higher rate. This finding corresponded to increasing construct expression levels, indicating that the functional impact of the longer isoform could be detected only at higher expression levels and longer timepoints (S7B–S7F Fig).

We also found that *SLK*-202 co-localizes with actin filaments more strongly than *SLK*-201 or the eGFP-only control (Figs 7E and S8A–S8D). At 48h, the *SLK*-202 transfected cells begin to lose their structure, and by 72h, the cells have mostly detached. Since the N-terminus of *SLK* contributes mainly to the cell death [57], we removed the N-terminal 373aa of *SLK*-202 ($\Delta_{1\text{-}373}$*SLK*-202). $\Delta_{1\text{-}373}$*SLK*-202 had a unique localization to actin filaments along the periphery of the cell (S8A–S8C and S8E Fig). $\Delta_{1\text{-}373}$*SLK*-201 looked similar to *SLK*-201. These results suggest that *SLK*-202 localizes to and potentially disassembles actin more effectively in metastatic melanoma compared to *SLK*-201, consistent with *SLK*-202 being more apoptotic in our binned data (Fig 7D).

## Clustering on DIR identifies correlations with genomic subtype and tumor location

To identify similarities in metastatic samples based on isoform expression patterns, we clustered the samples (columns in Fig 8). Rather than clustering raw expression data, we determined which of the kinase isoforms was significantly upregulated or downregulated in each of the 367 metastatic samples (see Methods) relative to all primary tumor samples. This allowed us to address the simpler question of which isoforms are altered in which samples. To identify correlated patterns of upregulation or downregulation we also clustered the isoforms (rows in Fig 8).

Of the 3,040 protein coding kinase isoforms, 235 had significant altered expression in > 13% of metastatic tumor samples. Clustering this reduced dataset with the k-means elbow method identified 4 sample clusters and 4 isoform groups (S9 Fig). However, we found that using k-means with 5 isoform groups strengthened certain BP enrichment patterns. These 5x4 clusters are depicted in Fig 8. For each sample cluster, we tested enrichment for batch ID, region (skin/soft tissue, lymph node, and distant metastasis), and genomic subtype.

Notable enrichments in Cluster A (n = 55 samples) include the tissue location of skin/soft tissue cluster and *BRAF* hotspot mutations. Cluster B (n = 69 samples) was identified as a lymph node cluster with mild enrichment in triple WT samples. Distant metastases were depleted in both A and B clusters. Cluster C (n = 60 samples) had no region enrichment but was strongly enriched for *RAS* hotspot mutations (Fisher's exact test, p = 4.4e-4, odds = 2.9). Cluster D (n = 183 samples) stood out as a low expression cluster, which had expression largely similar to the primary tumor samples, with little upregulation of isoforms compared to other groups. Moreover, decreased expression of isoforms (shown in blue) occurred in many samples. This cluster was enriched for distant metastases.

The batch ID enrichment analysis identified batch A18 in Cluster C, suggesting batch effects could have influenced our results. To address this issue, we clustered only the 199 metastatic samples (54% of all such samples) in batch A18 (S10 Fig), originally found in groups A-D. We found four clusters comparable to the four described above, and Cluster 3 was still significantly enriched for *RAS* hotspot mutants (p = 0.032, odds = 2.1). Clustering all samples *not* in A18, originally present in groups A-D, also revealed 4 clusters and though genomic subtype was not available for most of these samples, Cluster C still had the highest enrichment for *RAS* mutants (p = 0.16, odds = 2.3). Thus, the *RAS* group enrichment appears to be independent of the batch. Cluster D in our main heatmap was enriched for batch A37, a smaller batch (n = 41 samples), considerably smaller than the cluster it was in.

We also compared the level of 3' bias and sample impurity in each cluster and found that Cluster B had low purity (median 42%) compared to the other three (median of 72%, 80%, and

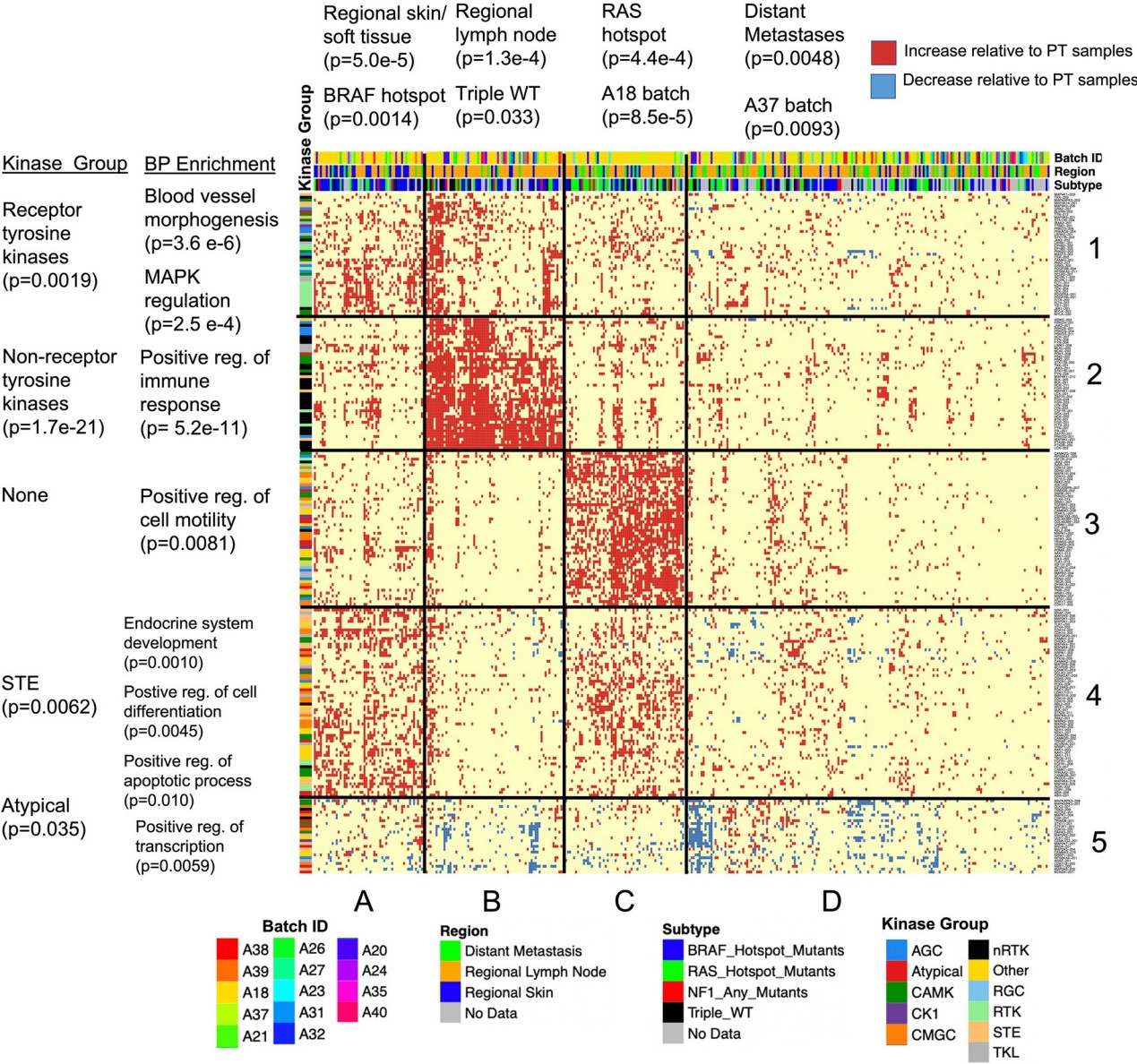

**Fig 8. Heatmap of 367 metastatic samples clustered according to kinase isoform counts.** Red dots indicate increased expression in metastases (Quasi-Poisson GLM, $p < 0.05$) while blue dots indicate decreased expression ($p < 0.2$). Shown are the 367 metastatic samples (columns) and 235 isoforms that were altered in $> 13\%$ of samples (rows). P-values for enrichments were calculated using the one-sided Fisher's exact test.

79% respectively). Median 3' bias did not differ noticeably, although Cluster C had a lowest mean bias (0.517, QoRTs score), indicating higher quality samples. Taken together, these data suggest that metastatic samples have characteristic subgroups related to tumor location and genomic subtype, where isoform expression patterns may help to identify the most similar samples to test as treatment subgroups.

## Isoform groups correlate with biological process annotations

We performed a similar analysis on the five isoform groups (i.e., rows), looking for kinase phylogenetic group and BP enrichments compared to the total human kinome. Group 1 was

enriched for genes involved in blood vessel morphogenesis (p = 3.6e-6) and related annotations, as well as MAPK regulation. These isoforms are upregulated in Clusters A and B. Since these genes are active in the skin/soft tissue sample cluster and regional lymph nodes, the isoforms may be important in the first transition from primary tumor to metastatic melanoma. This group is also enriched for RTKs.

Group 2 was strongly enriched for nRTKs and contained genes in the category of immune response, for example, used by leukocytes such as T-cells and B-cells (p = 5.2e-11). These isoforms are consistently upregulated in Cluster B. Due to their highly correlated expression and the low estimated purity of the Cluster B samples (median 42%), this group likely arises from immune cells infiltrating the tumor, consistent with previous findings from Akbani et al. [11]. Cluster B is also enriched for samples taken from lymph nodes, a prime location for immune cells to interact with the tumor.

Group 3 was enriched for kinases that regulate cell motility (p = 0.0081). No phylogenetic kinase group enrichments were found, although this group had weak CMGC enrichment compared to the other four groups in Fig 8. These isoforms had the highest expression in Cluster C, containing *RAS* hotspot mutant samples and distant metastases. We note a strong pattern of exclusivity for Group 3 isoforms with the immune infiltrate cluster of Group 2 isoforms, suggesting a novel means of stratifying samples for clinical testing.

Group 4 was enriched for kinases which positively regulate apoptosis (p = 0.010) and cell differentiation (p = 0.0045), and for STE kinases. These isoforms were upregulated in Clusters A and C. This group contains two isoforms of *CDK19*, a gene implicated in cancer proliferation (a third isoform, *CDK19-203*, lacks the seventh exon and decreases in metastatic samples). The function of these isoforms in apoptosis is not explored; on the one hand apoptotic processes may occur spontaneously in cancer due to cellular stress and DNA damage [58], on the other hand alternate splicing can modulate pro- and anti-apoptotic functions in the same gene, like *BCLX* [59]. Samples with high levels of immune infiltrate (i.e. Cluster B) appear to have no enrichment of these isoforms, indicating how therapeutics could be specific for one subgroup and be ineffective in another.

Group 5 contained isoforms of genes enriched for regulation of RNA biosynthesis and transcription (p = 0.0059). These isoforms had correlated downregulation in several samples (Clusters B and D), although they are not universally downregulated and in fact increase in some samples. One such gene, *NME1*, is a known suppressor of metastasis [60]. Also in this group are two isoforms of *MAPKAPK3* (-201 and -208), a gene which activates autophagy in response to stress [61] and represses transcription factor E47 [62]. A shorter isoform, -202, is increased in metastatic samples. This isoform lacks the p38 MAPK-binding site, meaning it cannot be activated by p38. This apparent isoform switching was not identified by our DIR analysis because isoform -201 increases in some metastatic samples. *RPS6KA4-201* also significantly decreases, though not the gene's two secondary isoforms -202 and -205. These isoforms lack a nuclear binding site on the 3' end, suggesting it is *RPS6KA4*'s nuclear binding that is selected against. The list of isoforms in Group 5 is given in Table 6, and the full list for each sample cluster and isoform group may be found in S8 Table.

Some isoforms had divergent expression patterns depending on cluster. For example, the major isoform of *BRD4*, *BRD4-201*, was found in Group 5, indicating decreased expression in several samples. In contrast, this isoform increased in *RAS*-mutant metastatic samples, as did two shorter isoforms *BRD4-205* (a member of Group 3) and *BRD4-203* (S11 Fig). This suggests *BRD4* may be a drug target specific to *RAS*-mutant melanoma; indeed, a recent study found that Vemurafenib-resistant melanoma was susceptible to *BRD4* degradation [63]. Consistent with this observation, DE analysis revealed an 11% increase in *BRD4* expression in metastatic *RAS* mutants, but this increase is not significant ($p_{unadjusted}$ = 0.452). Furthermore, we could

**Table 6. Kinase genes with correlated decreased expression in metastatic samples.**

| Identified by isoform clustering (Group 5)[1] | |
| --- | --- |
| **Gene** | **Isoform ID***  |
| MAP2K5 | 204 |
| MAPK7 | 204 |
| CAMKK2 | 216 |
| TNK2 | 208 |
| ARAF | 201 |
| MAP3K6 | 202 |
| MAP2K7 | 201 |
| MAPKAPK3 | 208, 201 |
| ULK1 | 201 |
| BCKDK | 201 |
| CSK | 201 |
| BRD4 | 201 |
| CDK16 | 220 |
| CDK11B | 203 |
| STK11 | 201 |
| ADCK2 | 201 |
| **RPS6KA4** | 201 |
| CLK3 | 201 |
| DAPK3 | 201 |
| CSNK1G2 | 201 |
| GTF2F1 | 201 |
| NRBP1 | 203 |
| TRIM28 | 201 |
| MAP2K2 | 202 |
| NME1 | 203 |
| MAP2K5 | 204 |
| MAPK7 | 204 |

These isoforms have correlated under-expression in individual samples, but not strong under-expression across all samples, with the exception of *RPS6KA4*, (FC = -1.39, $p_{adj}$ = 0.0011, *DESeq2* with high purity samples)

*isoform numbers are from Gencode v.29

not confirm *kallisto's* isoform assignments using exon junction alignment, although the reported increase in isoform 205 –a shortened isoform which includes the two bromodomains but not the C-terminal region or NET domain–may suggest an underlying switching effect.

## Immune infiltrate correlates with increased survival

In our analysis of survival across sample clusters, Cluster 2 (n = 67) was observed to have a higher median survival compared to the other three sample clusters (Fig 9A). Fittingly, this cluster corresponds to samples with immune infiltration. A log-rank test comparing Cluster 2 survival against the rest of the samples showed a low level of statistical significance (p = 0.065). Clusters 1 (n = 54), 3 (n = 60) and 4 (n = 175) demonstrated no significant difference in patient survival after applying pairwise log-rank tests.

We also analyzed the correlation between overall survival and HTSeq gene counts for each kinase gene. Of the 538 genes tested, *WNK2* and *OBSCN* presented the strongest negative correlation between expression (see Methods) and patient survival (Spearman $\rho$ = -0.26 and -0.24,

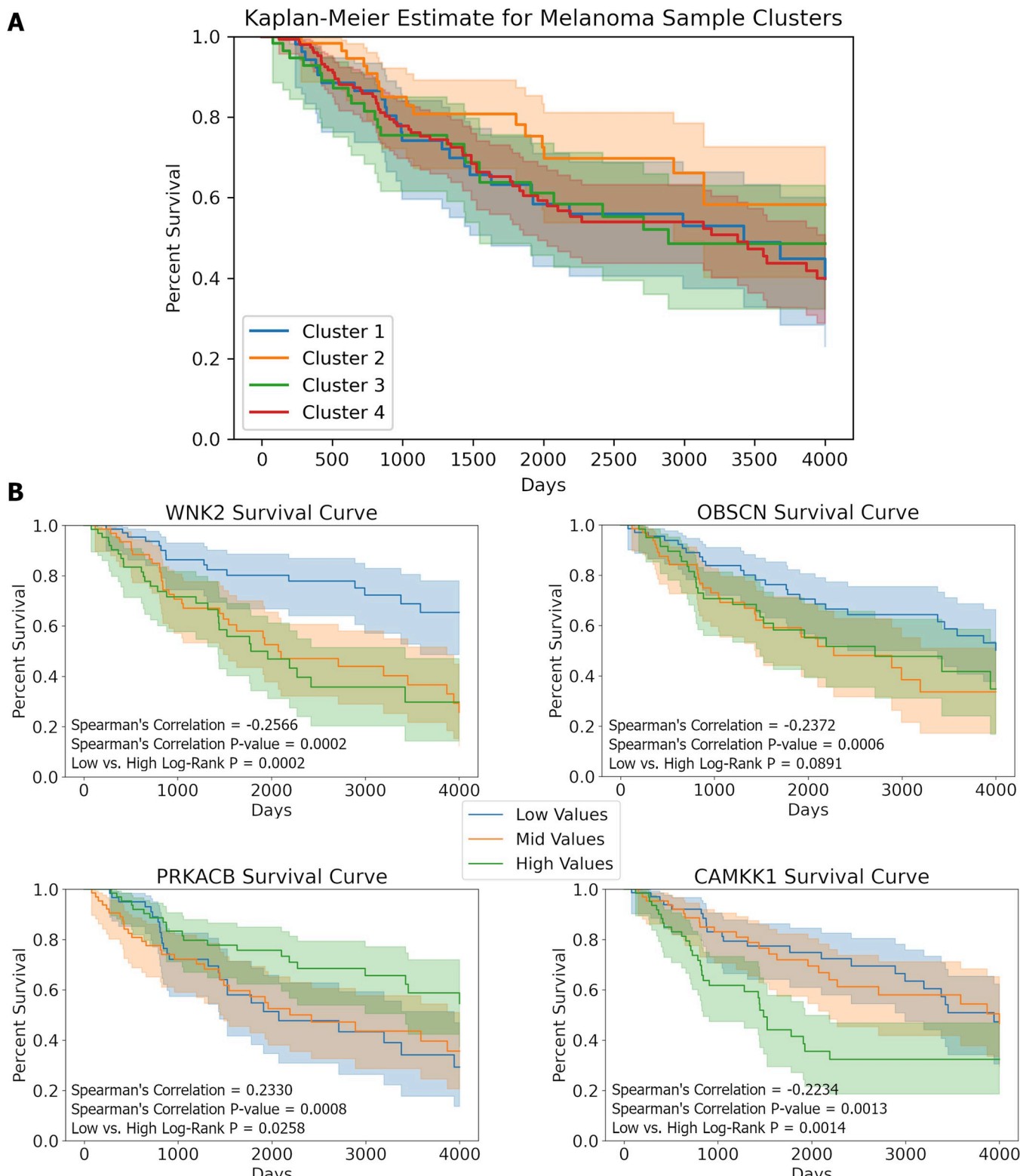

**Fig 9. Survival curves.** (A) Cluster 2 (corresponding to immune infiltrate) was observed to have higher median survival compared to the other clusters. A log-rank test comparing Cluster 2 survival against the rest of the samples showed a low level of statistical significance (p = 0.065). (B) Survival curves of the four genes with the highest Spearman correlation between gene expression and overall survival in high purity samples (n = 205). Samples were quantile binned by their gene count into low (blue), medium (red) and high (green) values. All survival curves were analyzed up to the 4000 days.

respectively), while *PRKACB* showed the strongest positive correlation ($\rho$ = 0.233)(Fig 9B). The unadjusted correlations were significant, however after multiple test correction (BH procedure), none of the correlation values rise to a level of statistical significance, with *WNK2* having the lowest adjusted p-value of 0.110.

## Discussion

Given the rise in melanoma cases across the world, and preliminary success of new therapeutic approaches combing kinase inhibitors and other treatments, we were encouraged to look for differential isoform expression, which has not been intensively studied, and compare it to differential expression identified using conventional approaches (i.e., using the gene locus as a proxy for average expression). We show that both differential expression and altered isoform ratios are prevalent in the human kinome in metastatic melanoma compared to primary tumor melanoma. Furthermore, these changes differ by genomic subtype and tumor location. Affected genes were enriched for several biological processes including immune response, angiogenesis, cell differentiation, chemotaxis, and cell projection organization. Our results provide insight into the regulation of melanoma progression and possible new routes for grouping therapeutic targets.

Different genes were affected by differential expression (DE) and differential isoform ratios (DIR). These genes differed in both phylogenetic groups, e.g. receptor tyrosine kinases in DE vs non-receptor tyrosine kinases in DIR, and biological process enrichments. Thus, isoform analysis may reveal novel information about cancer progression that DE analysis cannot. The drivers behind these splicing events are unknown, but can be multifactorial. For example, mutations in splicing factors can determine outcomes of alternative splicing, but so may somatic mutations or SNPs [64]. Additional determinants derive from epigenetic changes such as aberrant DNA methylation [65] and RNA modifications [66].

### Isoform switching may affect protein function

We chose to examine six genes with especially significant isoform switching in greater detail. Metastatic samples showed *SLK* overexpression in our study, something that has been previously observed in other cancer types such as *ErbB2*-driven breast cancer [67]. Knocking down this gene markedly reduces cell migration in *3T3 MEF* cells [68]. It appears that invasion is the functional benefit provided by SLK overexpression to metastatic melanoma. However, while the short form of *SLK* (*SLK-201*) is overexpressed in metastatic samples, the long form (*SLK-202*) is underexpressed. Overexpression of *SLK* can cause dimerization via the C-terminal coiled-coiled domain; these dimers then activate apoptosis [54]. The short form of *SLK* (*SLK-201*) skips an exon that encodes a coiled-coil region in the C-terminal domain. Our experiment found introduction of the *SLK-202* isoform to be more apoptotic at high concentrations; it is therefore possible that the decrease in the long *SLK*-202 isoform, seen in TCGA metastatic samples, decreases apoptotic potency and facilitates the transition toward metastasis. Thus, *SLK*-202 isoform expression may provide a therapeutic target. Furthermore, the transfected *SLK-202* isoform localized to actin filaments along the nuclear periphery more readily than the *SLK-201* isoform. Further experiments are needed to address the impact of the differential localization.

*MAP3K3* has been identified as an oncogene in various cancers [69–71]. Although we observed no differential expression of the gene (after immune-infiltrate samples were removed), we found that skipping of exon 3 was significant in metastatic samples. The functional effect of this skipping is unknown; it precedes, but is not part of, the PB1 protein-protein interaction domain. *MAP3K3* plays important roles in angiogenesis, cell differentiation, and proliferation and may regulate its partners through this structural edit.

In metastatic samples, *MKNK2* was found to switch to a shortened terminal exon which lacks the MAPK binding site. This switching has been previously observed in glioblastoma [32] (compared to normal samples), where the short terminal exon showed pro-oncogenic activity. The authors demonstrated that use of splice switching oligos in glioblastoma reduced the presence of the short terminal isoform and inhibited the oncogenic properties, suggesting this approach might also work in melanoma.

Another event we observed was in the *FES* gene, a non-receptor tyrosine kinase. The 11[th] exon, which encodes the SH2 domain and is necessary to activate the kinase domain [55], was skipped at a significantly higher rate in metastatic samples. *FES* has been previously identified as a tumor suppressor in melanoma [72], but we did not observe significant DE in our analysis. We predict that the skipping of the SH2 domain effectively turns off the kinase activity without decreasing the overall gene count. This effect would be consistent with reports of wild type *FES* acting as a tumor suppressor [73]. DE analysis alone would have missed this important effect. Notably, *FES* has several known inhibitors that target the SH2 domain and thus would not be effective against the short isoform [73].

*FGFR3*, which has highly significant negative DE, also has a significant alternative splicing event which affects the third Ig-like domain. There was a comparatively higher level of isoform *FGFR-205* (also known as *FGFR3-IIIc*) and less of *FGFR-202* (or *FGFR3-IIIb*). This *IIIb/c* imbalance has been observed in other cancers, such as colorectal [74]. The same study found that knocking down *FGFR3-IIIc* inhibited cell growth and induced apoptosis, but not *FGFR3-IIIb*. The negative DE was unexpected given *FGFR3* is often considered an oncogene, but the gene is known to limit growth in tumors of epithelial origin [75]. Hence the decreased expression of *IIIb* and switching to *IIIc* may be two separate mechanisms of altering *FGFR3* activity.

Finally, an isoform of *LIMK1* with an abrogated N-terminal LIM domain was expressed at a significantly higher level in metastatic samples. Deleting both LIM domains was previously found to increase kinase activity 3–7 fold [76], suggesting this isoform has greater kinase activity. Targeting *LIMK1* with small molecular inhibitors has been shown to reduce migration and invasion of malignant melanoma [56], suggesting increased activity would promote malignancy. *LIMK1* also did not have significant DE in our dataset.

## Expression pattern of *RAS* hotspot mutants

Our various analyses discovered that *RAS* mutants have an expression level pattern distinct from the other three genomic subtypes. *BRAF* and *MEK* inhibitors, while useful for treating *BRAF*-mutant melanoma, have no or limited effectiveness against *RAS* mutants [77]. *BRAF* mutants that gain resistance to *BRAF* inhibitors often acquire a secondary *NRAS* mutation [78], meaning any effective *RAS* mutant treatment may also aid in treating drug-resistant *BRAF*-mutants. We found that DE of kinases in *RAS* mutants is concentrated in CMGC kinases (as opposed to receptors as in the other three subtypes) and that DIR is concentrated in kinases involved in angiogenesis. Thus anti-angiogenics [79] are also possible treatments. Analysis of *kallisto* counts also identified the bromodomains of *BRD4* as a possible target. A recent study found that Vemurafenib-resistant melanoma was susceptible to *BRD4* degradation [63], and bromodomain inhibitors such as OTX015 and BI-2536 have already had some success in treating carcinomas [80]. However, this result was not supported by the *HTSeq* gene counts or exon junction analysis.

Another genomic subtype, triple WT melanoma, had DE mostly affecting $Ca^{2+}$/calmodulin-dependent protein kinase (in addition to RTKs), and we found *CAMKK1* expression had a negative correlation with survival (Fig 9B). These may also serve as a new set of drug targets for this rarer subtype.

## Further biological implications

One interesting result from the clustering analysis was the apparent mutual exclusivity of some kinase clusters in metastatic tumors. In particular, the isoform group involved in cell motility (i.e., Group 3) only had high expression in samples lacking in immune response markers (i.e., Cluster C). It is possible samples with this expression pattern, which includes many *RAS* mutants, may evade immune detection, which would explain this apparent mutual exclusivity. But it is also possible the low purity of these samples obscures increased expression of Group 3. Additionally, cell differentiation and apoptotic markers were highly expressed in regional soft tissue tumors (i.e., Cluster A) and *RAS* mutants (Cluster C), but not lymph node tumors (i.e., Cluster B). $BRAF^{V600E}$ mutations are present in Clusters A and B, indicating that in addition to the driver mutation, location of the tumor and isoform content is relevant to discern tumor biology and treatment choices. We conclude that the heterogeneity of sample types displayed in Clusters A-D suggests that the complexity of tumor biology is greater than indicated by driver mutations alone, and that the isoforms in our heatmap may be useful for screening metastatic samples.

## Limitations

The present study has limitations that may impact the interpretations of our data. For example, isoform count estimation is a computational approach to predict isoform expression levels from short read data. Other short read algorithms–using direct alignment approaches such as *RSEM*, *Sailfish*, or *Cufflinks*–may produce different count estimates than *kallisto*. The accuracy of these algorithms decreases as the number of gene isoforms increases. However, one study found that for genes with <15 isoforms, *kallisto* estimated counts still had >0.95 correlation with simulated "ground truth" counts, excluding very short transcripts [81]. Tested genes in our study had a median of 5 and mean of 6.3 coding isoforms. Nonetheless, we also analyze reads aligned to exonic junctions to verify *kallisto* findings.

Because *kallisto* requires isoform transcript sequences, our method does not account for novel isoforms. Specialized tools exist for this, such as *psiCLASS* [82], but this was not the focus of the present study. Here we rely on a fast isoform quantification that relies on an existing genome annotation. We compared our method to a standard approach, *DEXSeq*, which performs local exon analysis based on the architecture of *DESeq2*. Our method proved more sensitive to exon splicing events and is computationally faster than *DEXSeq* for hundreds of samples. 3$^{rd}$-gen RNA sequencing technologies such as PacBio [22] and Oxford Nanopore [23] are anticipated to provide more accurate knowledge of isoform sequences, both annotated and novel.

Sample artefacts could also affect our results. As indicated by the results presented, computational estimates of isoform counts are highly impacted by sample impurity or 3' fragment bias. We removed problem samples in our study to obtain higher confidence results. Although our quality-controlled sample set had little difference in purity between primary tumor and metastatic samples (two-sided Wilcoxon, p = 0.88), primary tumor samples still exhibited increased 3' bias compared to metastatic tumors (p = 4.4e-4). Estimates of fragment bias could be incorporated into the existing tools to reduce artefactual results.

With one exception, the melanoma TCGA samples are not matched, i.e. the primary tumor and metastatic samples do not come from the same patient. However, our sample size is large enough to make meaningful comparisons between sample categories.

## Summary

We have compared differential gene expression and differential isoform expression to address the hidden effect of differential splicing of kinases in metastatic melanoma. We demonstrate

novel, plausible stratification of tumors for clinical testing, for example, immune infiltrate vs. cell migration groups. These groups are consistent with presence of a specific driver mutation (i.e., *BRAF*^(V600E)), but a mixture of samples could be found in each group. Additionally, we identified a group of isoforms with significant downregulation in metastatic tumors. These include a known suppressor of metastasis (*NME1*), and may provide a rich source of discovery for additional suppressors. Although we focused here on the kinome in metastatic melanoma, in future work we can expand the analysis to the entire human genome, as well as other cancer types having a rich source of expression data. Further experimental work can confirm links between isoform switching and angiogenesis or other cell processes.

## Supporting information

**S1 Fig. Differential protein expression analysis results for the high purity SKCM samples.** Wilcoxon's rank-sum test was performed on 208 protein probes between 78 primary and 146 metastatic high purity tumor samples. Shown are 14 kinases that were significant at the level of BH adjusted p-value<0.05. X-axis labels indicate the RPPA probe and the corresponding gene encoding that protein.
(TIF)

**S2 Fig. Fragment bias in primary tumor samples drives significance in *EIF2AK4*. (A)** Two protein coding isoforms of *EIF2AK4*, the full-length isoform (-201) and 3' fragment (-205). **(B)** Change in DIR significance ($-\log_{10}$p) as samples are removed one-by-one in order of highest bias (red and orange dots) vs in order of lowest bias (green or blue dots). The significance drops faster when the high-bias samples are removed. The p-value here is calculated using the PCA method with the coin general independence test. **(C)** When only primary tumor samples are removed, the differences in p-values are even more disparate, indicating that high-bias primary tumor samples drive significance. **(D)** Box plots for the three isoforms with the highest number of normalized counts. Significance is driven by a higher amount of the full-length isoform in metastatic samples but a lower amount (on average) of the 3' fragment. **(E)** Scatter plot of the raw counts of each isoform in each sample. Circled in black are the ten isoforms with the highest 3' bias, indicated by high levels of the 3' fragment and low levels of the full-length isoform.
(TIF)

**S3 Fig. Quality control reduces DIR significance of *PTK2B*.** Expression of isoform *PTK2B-205* in particular is driven by low-purity metastatic samples. Its expression drastically decreases when they are removed. Conversely, there is lower average expression of *PTK2B-203* in primary tumor samples before samples with high 3' bias are removed. This is likely due to the presence of more exons on the 5' end, which will be undercounted in samples with 3' bias.
(TIF)

**S4 Fig. Differential isoform ratios in FGFR3.** Plotted are **(A)** TPM counts and **(B)** fraction of all isoform counts for each sample. Although the trend is decreased expression, one isoform (*FGFR3-205*) has mildly increased expression, resulting in highly altered isoform ratios.
(TIF)

**S5 Fig. *DEXSeq* Results. (A-D)** *DEXSeq* confirmed DIR in three genes: *MAST4*, *FGFR3*, and *SLK*. The alternate promoter of *LIMK1* was also significant before p-value adjustment. **(E-F)** The 3^(rd) exon of *MAP3K3* (bin 9) and MAPK-binding region of *MKNK* (bin 4) did not test significant with DEXSeq, even before p-value adjustment, despite testing as significant using exon junction alignment. **(G-H)** The 14^(th) bin of *LMTK3* and 1^(st) bin of *PDGFRA* also tested as

highly significant. However, these two exons have low expression (median ~1 count) so this result is likely due to noise and is unlikely to have biological relevance.
(TIF)

**S6 Fig. SLK isoform expression induces apoptosis. A)** Plot of uniquely mapping sequence reads for A375 cells showing skipping of *SLK* exon 13. Original RNA-seq data are from the Sequence Read Archive SRR961660, https://www.refine.bio/samples/SRR961660. **B)** A bar graph showing annexin V staining over the 72h time course for 2 biological replicates. We see an increase in percent annexin V for both *SLK* isoforms at 48h and 72h compared to the eGFP-only control. All significant t-tests (*) had p-values < 0.05. All non-significant (NS) t-tests had p-values > 0.05. T-tests for the negative control were not included on the graph.
(TIF)

**S7 Fig. Analysis of annexin V staining in cells expressing GFP-fusions of *SLK* isoforms after binning cells. A)** Illustration of thresholds used for determining annexin V positive cells (determined experimentally for each replicate, see *Methods*) in GFP-expressing cells (> 10^4 fluorescence units). Bins B1 through B5 represent cells with increasing GFP expression, and therefore also increasing levels of corresponding *SLK* isoform. Bin B5 is larger to accommodate the reduced number of cells expressing high levels of GFP. **B-E)** Comparison of incidence of annexin V positive cells in different constructs across different bins of increasing GFP expression (B1 –lowest, B5 –highest); $^{**}$(p < 0.01), $^{*}$(p < 0.05), $\nabla$(p < 0.056).
(TIF)

**S8 Fig. Microscopy of *SLK* isoform expression and actin localization.** Representative images of each construct (green) at the **A)** 24-hour, **B)** 48-hour, and **C)** 72-hour timepoints. At each timepoint, cells were stained with DAPI (blue) and phalloidin (red). **D)** Merged channel images of eGFP-only, *SLK*-202-eGFP, and *SLK*-201-eGFP over the time course experiment. **E)** Merged channel images of $\triangle_{1-373}$SLK-202 and $\triangle_{1-373}$SLK-201 over the time course experiment.
(TIF)

**S9 Fig. Heatmap of 367 metastatic samples clustered (4x4) according to kinase isoform counts.** Red dots indicate increased expression in metastases (Quasi-Poisson GLM, p<0.05) while blue dots indicate decreased expression (p<0.2). Shown are the 367 metastatic samples (columns) and 235 isoforms that were altered in >13% of samples (rows). P-values were calculated using Fisher's exact test.
(TIF)

**S10 Fig. Clustering of batch A18 and non-batch A18 samples.** Isoform groups are the same as in Fig 8. Both sample subsets separated into four clusters comparable to Fig 8.
(TIF)

**S11 Fig. Heightened expression of *BRD4* isoforms in RAS mutant metastatic samples.** Although total *BRD4* counts did not test as having significant DE between any group of primary and metastatic tumors, isoforms *BRD4-203* and *BRD4-205* have heightened expression in *RAS*-mutant metastatic samples. Exon junction analysis could not confirm these particular isoforms from sequence reads.
(TIF)

**S1 Table. DESeq2 Results.**
(XLSX)

**S2 Table. Differential expression of RTKs using two *DESeq2* models.**
(TIF)

**S3 Table. Sample Metadata.**
(XLSX)

**S4 Table. BP Enrichments.**
(XLSX)

**S5 Table. RPPA data.**
(XLSX)

**S6 Table. DIR results.**
(XLSX)

**S7 Table. DEXSeq Results.**
(XLSX)

**S8 Table. Heatmap Clusters.**
(XLSX)

**S1 Results. Supplemental results file showing extended analyses.**
(DOCX)

## Acknowledgments

We would like to thank Stacie Anderson and Martha Kirby of the NHGRI Flow Cytometry Core for helping with the design and execution of the annexin V experiments. Additional thanks to Jan Wisniewski, Ph.D. the Optical Microscopy Laboratory at the National Cancer Institute, the National Institutes of Health for assisting us with imaging and image processing for the actin disassembly experiments.

## Author Contributions

**Conceptualization:** Laura Elnitski.

**Formal analysis:** David O. Holland, Valer Gotea, Catherine Baugher, Hua Tan.

**Funding acquisition:** Laura Elnitski.

**Investigation:** David O. Holland, Kevin Fedkenheuer, Sushil K. Jaiswal.

**Methodology:** David O. Holland, Valer Gotea.

**Software:** David O. Holland, Valer Gotea.

**Supervision:** Laura Elnitski.

**Validation:** David O. Holland, Valer Gotea, Kevin Fedkenheuer, Sushil K. Jaiswal, Michael Fedkenheuer.

**Visualization:** David O. Holland, Valer Gotea, Kevin Fedkenheuer, Sushil K. Jaiswal, Catherine Baugher, Hua Tan, Michael Fedkenheuer.

**Writing – original draft:** David O. Holland.

**Writing – review & editing:** David O. Holland, Valer Gotea, Kevin Fedkenheuer, Sushil K. Jaiswal, Catherine Baugher, Hua Tan, Laura Elnitski.

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
