## [Decision Letter · Decision Letter 0]

5 Jul 2021

Dear Dr. Elnitski,

Thank you very much for submitting your manuscript "Characterization and clustering of kinase isoform expression in metastatic melanoma" for consideration at PLOS Computational Biology.

As with all papers reviewed by the journal, your manuscript was reviewed by members of the editorial board and by several independent reviewers. In light of the reviews (below this email), we would like to invite the resubmission of a significantly-revised version that takes into account the reviewers' comments.

There are important suggestions from all reviewers as they highlight important aspects to consider to clarify how the analysis was done, ensure that the manuscript follows gold standard methods in the field (particularly from reviewer 1), incorporate additional cancer genomics datasets that are publicly available, and aim to look deeper into the experimental validation of the computational predictions.

We cannot make any decision about publication until we have seen the revised manuscript and your response to the reviewers' comments. Your revised manuscript is also likely to be sent to reviewers for further evaluation.

Sincerely,

Pau Creixell

Guest Editor

PLOS Computational Biology

Jian Ma

Deputy Editor

PLOS Computational Biology

Reviewer's Responses to Questions

**Comments to the Authors:**

**Reviewer #1:** This manuscript uses gene expression analyses to address the hypothesis that differential expression of kinase genes and isoforms defines differences between normal melanocytes, primary and metastatic melanoma. Focusing on the kinase-encoding genes and isoforms present in the Gencode annotation, the authors performed a comparison between TCGA cohorts including primary and metastatic melanomas. After filtering for immune infiltrates and impurities in the tumor sample, just under 200 genes were found to be statistically significantly changed. There were some differences between genomic subtypes in the subclasses of kinase enrichment. Gene ontology analysis required division into genomic subtypes to reveal enrichment of distinct biological processes. Correcting for 3’ bias, they next examined differential isoform expression, the highest confidence group exhibited 60 genes with differential isoform expression between primary and metastatic tumors. A small number of high-confidence isoform differences were then examined for their effects on protein domain composition. From this analysis, the hypothesis that one isoform of SLK versus another would affect apoptosis frequency was tested, but both isoforms appeared to be equally pro-apoptotic. Finally, samples were clustered based on differential isoform expression to reveal several groups with coordinated expression changes, and the biological implications of these clusters were discussed.

The focus of the manuscript is well placed, choosing to ‘zoom in’ first on the kinome and more specifically on isoforms that change during transitions between normal (to the small sample size possible) to primary to metastasis in the context of metastatic melanoma enables a reasonably sized ‘omics’ space in which to derive meaningful analyses. The computational and statistical implementation itself is well done in its own right, and the manuscript addresses problems of interest to a broad cancer biology and gene expression audience.

My primary criticism is the choice to approach the isoform variation problem, when the data consists of short reads, with a full-length isoform estimation process. This approach also reveals a deficit of familiarity with the splicing literature—for analysis of short read data, differential isoform analysis is typically done by examining local splicing/promoter/alternative last exon variations within a gene, rather than trying to differentiate full-length transcript isoforms. Rather than trying to reinvent the wheel, the authors should perform such local splicing analysis using a number of software packages that were developed specifically for the purpose (e.g, DEXseq, rMATs, MAJIQ, MISO). In fact, DEXseq uses the same architecture that they have used here for total gene expression analysis, DESeq2. Artifacts such as 3’ bias, which the authors correctly consider here in their analyses of full-length isoforms, are less of a problem in local splice variation, since the exons within a small segment of a transcript are subject to similar positional biases. These approaches also allow for detection of novel splice variants in the tumor samples which as the authors correctly note may be unaccounted for in the annotation used here--the authors seem to be unaware of the literature post-2011 in stating “Currently, novel isoforms may be indirectly inferred by aligning reads to the nucleotide sequences of individual exons [72].”

As stated in the discussion, “accuracy decreases as the number of gene isoforms increases”. This is because short reads from common regions of the isoforms of a single gene must be algorithmically distributed among the variants, which cannot be defined discretely in the absence of full-length transcript annotation. Thus it is likely that many reads are allocated to non-existent isoforms and removed from the quantification of existing isoforms--further reason to switch to local analysis. It is very probable that a higher number of differentially expressed isoforms will be identified using one of the dedicated analysis packages, resulting in different outcomes—for example, the different gene profile in DE vs DIR could be a result of not detecting most of the alternative isoforms that are actually present, thereby underestimating the overlap.

There is a section of the paper that is heavily oriented toward gene ontology, but even as the authors acknowledge the limitations of these analyses, there is much made of the gene groups identified. The manuscript would benefit from tailored gene ontology emphasizing the known functions of the kinase genes. Heavy reliance on gene ontology annotation (the groups are only as good as the annotations that produce them) may obscure other relevant biological processes. For example, what are all of the genes DE in “Ephrin receptor signaling pathway”, and are they all real examples of this pathway if you look at them individually?

What is the relevance of nuclear trafficked kinases? This is never discussed.

The SLK overexpression experiment lacks controls. It is necessary to perform a western blot to determine expression levels of the two isoforms. Are the two isoforms distinguishable on blot? If so, is the endogenous protein ratio shifted in response to one or the other isoform overexpression? If not, looking at the RNA level could determine whether the endogenous splice variants are affected by overexpression. The endogenous pool could re-equilibrate the ratios between isoforms, obscuring the differences. Further, the assay used here to determine apoptosis is not specific to apoptosis, it is a general cell viability assay. Better to use something like Annexin V staining. More importantly, this experiment would be much more quantitative if done with a stable, inducible transgene (e.g. doxycycline) so as to remove the differences across the population due to variable transfection uptake and expression.

There are a few instances of over-interpretation of data, e.g.:

Pg 18, “this result suggests a distinct set of alterations are necessary for RAS hotspot mutants to become metastatic”; rather a distinct set of alterations is associated with RAS hotspot mutations in metastatic tumors.

A good portion of the discussion is highly speculative given the data presented here—for example, “The presence of distant metastases in Clusters C and D, which have little immune marker expression, suggest they may have evaded the initial immune attack in the lymph nodes” and “Perhaps cells that reach the lymph node have already evaded apoptosis.” These observations may suggest hypotheses to be further tested, but statements such as these represent multiple experimental steps beyond the data in question.

**Reviewer #2:** Holland et. al. studied in great detail the differences in gene and isoform expression of kinases between primary and metastatic melanoma samples from The Cancer Genome Atlas (TCGA). The paper is very thorough in all the analyses and addresses an important topic: the vast majority of cancer-related deaths are due to metastases, not primary tumors, and the differences between the two stages are not sufficiently studied.

Major points:

1 - My main concern with the paper is that it is very difficult to follow which samples are being studied at each point, as the authors keep changing the dataset: from the complete dataset, to excluding samples due to low purity, over-representation of 3' ends or batch effects.

While the reasoning behind each of these changes is scientifically sound and well-justified, it makes comparisons across experiments very difficult to follow.

For example, the authors make a good case for excluding some samples in the gene-expression analysis due to having low purity and this leading to biases caused by immune-infiltrate and not changes in the cancer cells. But then, they go back to study the entire dataset for the first differential isoform analysis, and then excluding some other samples due to other biases, making the comparison between the results of the isoform changes and differential expression very difficult to follow for the reader.

This could be solved, for example, by discussing the results of the DE and DIR experiments, with the same cohorts, side-by-side (i.e. the ones with the complete dataset, then the ones excluding the samples with low purity, then by hotspots etc.)

Minor points:

1 - Have the authors considered using other orthogonal data from TCGA to validate some of the findings? I am thinking particularly about the SLK hypothesis. I think it is great that the authors have tried to validate it in cell lines, and also that the negative outcome of the experiment should be reported, but maybe they could use gene expression or RPPA data to test whether the apoptosis pathway is more or less active depending on the SLK isoform being expressed.

**Reviewer #3: **In this study by Holland et al, they perform a comprehensive analysis of differentially expressed (DE) kinases as well as differentially expressed kinase isoforms (based on ratios - DIR) in primary and metastatic melanoma. This research and the findings should be of interest to a broad audience. The authors uncover that DIR is likely to play an important role as cells progress to a metastatic state and the differential expression play an important role in this process as well and clusters based on genetic drivers such as BRAF V600 mutations vs NRAS mutations. The research can point to novel therapeutic targets that become over expressed as cells progress to the metastatic state and therefore provides insights into potential targets in metastatic melanoma.

Minor Comments:

1. TCGA studies now incorporate RPPA (reverse phase protein arrays) analysis which will provide data on which proteins have increased expression. Could the author incorporate the protein expression analysis with the differential expression analysis to see how often differential expression correlates with increased protein expression. Furthermore these may provide a better snap shot of bona fide targets where increased expression correlates with detected increased protein expression for a subset of kinases. Are RTKs also over represented in RPPA analysis of metastatic samples?

2. TCGA studies also incorporate survival data and is there a signature for metastatic tumors that may correlate with better or worse prognosis? Ie does increased expression of an RTK cluster portend poorer survival? Does decreased expression of RTKs predict better survival for example?

3. It is very interesting the differential expression of RTKs is observed for the BRAF mutant subset of melanomas. In general V600 mutant melanomas often suppress signaling through other survival pathways to avoid cell death that can happen upon hyperactivation of numerous cellular proliferation and survival pathways. Inhibition of mutant BRAF relieves the suppression and often results in activation of upstream signaling cascades such as EGFR. Evaluation of Table S2 indicates there is a decrease in EGFR expression, which would be consistent with this observation. However, numerous RTKs have increased expression such as AXL. How does this correlate with mutant BRAF expression in BRAF mutant melanomas? Is mutant BRAF expression decreased? Having increased expression of known drivers such as MERTK in metastatic melanoma or RTKs that drive resistance to RAF inhibitors is very exciting and could drive therapeutic intervention strategies in the clinic – the authors should discuss the implications of these findings.

4. For the SLK experiments, it would be informative to know if this is apoptosis caused by the two different SLK isoforms (sub-2n DNA content by FACS analysis for example). Furthermore it would be interesting if the SLK isoform lacking the coiled-coil domains actually promotes cell proliferation or migration for example, knocking out this isoform in a cell line expressing this shorter isoform would be informative.

**Have the authors made all data and (if applicable) computational code underlying the findings in their manuscript fully available?**

Reviewer #1: Yes

Reviewer #2: Yes

Reviewer #3: Yes

PLOS authors have the option to publish the peer review history of their article (what does this mean?). If published, this will include your full peer review and any attached files.

Reviewer #1: No

Reviewer #2: No

Reviewer #3: No
---

## [Decision Letter · Decision Letter 1]

27 Nov 2021

Dear Dr. Elnitski,

Thank you very much for submitting your manuscript "Characterization and clustering of kinase isoform expression in metastatic melanoma" for consideration at PLOS Computational Biology. As with all papers reviewed by the journal, your manuscript was reviewed by members of the editorial board and by several independent reviewers. The reviewers appreciated the attention to an important topic. Based on the reviews, we are likely to accept this manuscript for publication, providing that you modify the manuscript according to the review recommendations.

Please note that reviewers 2 and 3 are fully satisfied with the revised version of the manuscript, while reviewer 1's concerns with the experiments in figure S6 are not fully resolved, and proposes an approach to address them. We would like you to address these concerns before we can reach a final decision on the manuscript.

Sincerely,

Pau Creixell

Guest Editor

PLOS Computational Biology

Jian Ma

Deputy Editor

PLOS Computational Biology

[LINK]

Reviewer's Responses to Questions

**Comments to the Authors:**

Reviewer #1: I am satisfied that the authors have properly addressed many of my criticisms concerning their differential splicing analyses. To the extent that differential isoform expression can potentially stratify tumors based on genomic subtypes, metastatic potential, and even prognostic indicators is a key finding.

The results of the sole experimental portion of the manuscript (SLK isoform effects on apoptosis) are disappointing, given that the underlying hypothesis of the study is that splicing isoforms can affect biological processes differentially. I wonder whether this experiment is being analyzed in the most appropriate way. In Fig S6 B, the FACS plots are shown for the eGFP control versus negative. The FACS plots for the two SLK-GFP fusions are not shown. If there is a difference in the function of the two SLK isoforms at different concentrations, it should be possible to compare the percentage of Annexin-positive cells within SLK-GFP expression bins (divided along the x-axis) between the two constructs. The prediction would be that SLK-202 would exhibit higher Annexin staining within lower GFP expression bins than SLK-201.

Without the isoform function data, the question is whether the differential isoform expression signatures constitute a significant advancement in classification/prognostication for melanomas.

Reviewer #2: The authors have addressed all my concerns

Reviewer #3: The authors have comprehensively addressed all of my concerns and I feel the manuscript should now be accepted for publication in PLOS Computational Biology.

**Have the authors made all data and (if applicable) computational code underlying the findings in their manuscript fully available?**

Reviewer #1: Yes

Reviewer #2: Yes

Reviewer #3: Yes

PLOS authors have the option to publish the peer review history of their article (what does this mean?). If published, this will include your full peer review and any attached files.

Reviewer #1: No

Reviewer #2: No

Reviewer #3: No

Figure Files:

Data Requirements:

Reproducibility:

References:

---

## [Decision Letter · Decision Letter 2]

29 Mar 2022

Dear Dr. Elnitski,

We are pleased to inform you that your manuscript 'Characterization and clustering of kinase isoform expression in metastatic melanoma' has been provisionally accepted for publication in PLOS Computational Biology.

Best regards,

Pau Creixell

Guest Editor

PLOS Computational Biology

Jian Ma

Deputy Editor

PLOS Computational Biology

Reviewer's Responses to Questions

**Comments to the Authors:**

Reviewer #1: The new analysis of the FACS data supports the hypothesis that the SLK isoform that is more highly expressed in metastatic compared to primary tumors is a less potent inducer of apoptosis when expressed at high levels. These data provide an example of the biological relevance of the splicing changes observed in the metastatic kinome and enhance the descriptive data in this manuscript with experimental evidence.

**Have the authors made all data and (if applicable) computational code underlying the findings in their manuscript fully available?**

Reviewer #1: Yes

PLOS authors have the option to publish the peer review history of their article (what does this mean?). If published, this will include your full peer review and any attached files.

Reviewer #1: No

---

## [Editor Report · Acceptance letter]

29 Apr 2022

PCOMPBIOL-D-21-00725R2 

Characterization and clustering of kinase isoform expression in metastatic melanoma

Dear Dr Elnitski,

I am pleased to inform you that your manuscript has been formally accepted for publication in PLOS Computational Biology. Your manuscript is now with our production department and you will be notified of the publication date in due course.

With kind regards,

Agnes Pap
